# PredLDM: Spatiotemporal Sequence Prediction with Latent Diffusion Models

**Yechao Xu**                                                             *zhigaoqishi@qq.com*
*State Key Laboratory for Novel Software Technology, Nanjing University*

**Zhengxing Sun**[*]                                                          *szx@nju.edu.cn*
*State Key Laboratory for Novel Software Technology, Nanjing University*

**Qian Li**                                                        *public_liqian@163.com*
*College of Meteorology and Oceanography, National University of Defense Technology*

**Jiao Qu**[*]                                                            *qujiao@nju.edu.cn*
*State Key Laboratory of Pharmaceutical Biotechnology and Nanjing Drum Tower Hospital the Affiliated Hospital of Nanjing University Medical School, School of Life Sciences, Nanjing University*

**Reviewed on OpenReview:** *https://openreview.net/forum?id=TWmnOUzcCo*

## Abstract

Predicting the accurate and realistic future is an attractive landmark in spatiotemporal sequence prediction. Despite recent progress in spatiotemporal predictive models, explorations in this field are challenging due to difficulties in intricate global coherence and comprehensive history understanding. In this study, we introduce latent diffusion models (LDMs) into spatiotemporal sequence prediction (PredLDM) with a two-stage training paradigm. (i) To compress intricate global coherent spatiotemporal content into latent space, we propose the masked-attention transformer-based variational autoencoder (MT-VAE) by exploiting transformers with masked self-attention layers. (ii) Different from LDMs in generation-related fields where the condition in our problem settings is historical observations instead of texts, the condition-aware LDM (CA-LDM) is provided for comprehensive understanding of historical sequences. Our denoising diffusion process learns the distribution of both conditional generation and condition-aware reconstruction. Results on KittiCaltech, KTH and SEVIR datasets show that our PredLDM provides promising performance and realistic predictions in multiple scenarios including car driving, humans and weather evolutions. (https://github.com/MaoWuToday/PredLDM.git)

## 1 Introduction

Spatiotemporal sequence prediction is a fundamental task in computer vision that given a sequence of images, neural networks predict the subsequent image sequence to describe what will happen in the future (Oprea et al., 2020; Shi et al., 2015). Different from video generation (Ho et al., 2022b) predicting from text prompts or unconditionally, this task is conditioned on historical observations of dynamic scenes (Oprea et al., 2020). By learning underlying spatiotemporal patterns from successive data with unsupervised manners, an ideal model is to predict accurate dynamics with realistic visual appearance (Lee et al., 2018). It can serve various disciplines, such as autonomous driving (Kwon & Park, 2019), robotics planning (Finn et al., 2016), traffic management (Liu et al., 2024b) and weather forecasting (Zhang et al., 2023b).

For producing future frames, classical predictive models are mostly optimized by minimizing mean error between predictions and ground truth across spatial and temporal dimensions (Oprea et al., 2020). Shi

---

[*]Corresponding authors

et al. (2015) introduce ConvLSTM networks, which is a milestone at grasping spatiotemporal aspects with convolutional recurrent architectures. Inspired by this, advanced recurrent models (Wang et al., 2017; 2018; Wu et al., 2021b; Sun et al., 2023; Villegas et al., 2017; Oliu et al., 2018) and recurrent-free ones (Gao et al., 2022; Tan et al., 2023) emerge out. However, the mean error-related objective leads to generating blur for uncertain future outcomes (Oprea et al., 2020; Lee et al., 2018). To improve visual quality, although generative models like variational autoencoders (VAEs) (Villegas et al., 2019; Wu et al., 2021a; Babaeizadeh et al., 2021), generative adversarial networks (GANs) (Clark et al., 2019; Tulyakov et al., 2018) and flow-based models (Dorkenwald et al., 2021) are alternatives, they are easy for mode collapse and the performance is not satisfactory. As LDMs reveal promising performance with high-fidelity appearance especially in T2I (Nichol et al., 2021; Rombach et al., 2022) and T2V (Ho et al., 2022b; He et al., 2022) through learning joint distributions with conditions in latent space by iterative denoising diffusion processes (Rombach et al., 2022), we introduce LDMs into spatiotemporal sequence prediction, under consideration of intricate global coherence and comprehensive history understanding.

In the light of intricate global coherence, existing predictive models are restricted by finite-scale temporal variations within training samples, whereas temporal transformations are complex and nearly infinite in nature. For simulating intricate temporal patterns, a solution is to model as many diverse variations as possible in the pretraining stage (Devlin, 2018; He et al., 2022) by masked modeling (Xie et al., 2022; Cheng et al., 2022). This can serve perceptual compression in LDMs (Singer et al., 2022). Meanwhile, existing LDMs exploit 3D convolutions (Ho et al., 2022b; He et al., 2022) and convolutional temporal layers (Singer et al., 2022) to extend T2I models to T2V applications. This leads to global dependencies being neglected, limited by compression of the local receptive field of convolutions (Li et al., 2023). For modeling global reliance, transformers are natural alternatives. To solve intricate global coherence, we expect to propose a transformer-based VAE with masked modeling. In another light of comprehensive history understanding, different from generation-purpose models conditioned by text prompts, the condition in this problem setting is historical image sequences. Compared to text prompts describing scenes with highly dimensional symbols, conditions of spatiotemporal sequences are more difficult for machines to understand as raw pixels are low-level and diverse. It is expected to leave conditions comprehensively understood in latent space during denoising diffusion processes.

With respect to these problems, we propose a spatiotemporal predictive model called PredLDM. (i) To compress intricate global coherent spatiotemporal content into latent space, we propose exploiting transformer-based VAE with masked attention to capture complex and global coherence in MT-VAE. (ii) To comprehensively understand the historical observations, condition-aware latent diffusion is performed. The denoising diffusion process of CA-LDM learns the distribution for both conditional generation and condition-aware reconstruction.

Extensive experiments are conducted on KittiCaltech (Geiger et al., 2013), KTH (Schuldt et al., 2004) and SEVIR (Veillette et al., 2020) datasets. Results show accurate performance and realistic visual appearance of trained PredLDM, indicating the promising future of this study. Contributions can be summarized as:

- To predict the accurate and realistic future image sequences, we propose a spatiotemporal predictive model called PredLDM, by introducing LMDs into this field under consideration of intricate global coherent modeling and comprehensive history understanding.

- For intricate global coherence, MT-VAE is proposed by transformers with masked attention variationally compressing complex temporal patterns and global reliance.

- Considering comprehensive history understanding, CA-LDM is performed by learning distributions of both conditional generation and condition-aware reconstruction.

- Experiments on several datasets including KittiCaltech, KTH and SEVIR show superior performance of PredLDM with realistic appearance, revealing potential for continuous research and applications.

## 2 Related Works

### 2.1 Spatiotemporal Sequence Prediction

Spatiotemporal sequence prediction produces the future sequence of images given by historical observations to describe what is going to happen (Oprea et al., 2020). This research direction originates from predictive coding (Huang & Rao, 2011; Rao & Ballard, 1999) which reveals the human behavior predicting visual signals through both space and time dimensions. Initial attempts from Ranzato et al. (2014) and Srivastava et al. (2015) introduce recurrent language baselines to model natural spatiotemporal signals. For explicitly modeling spatial information, Shi et al. (2015) propose using convolutions to replace fully connected layers in recurrent units. This attempt greatly inspires the progress on the recurrent predictive architectures (Wang et al., 2017; 2018; Wu et al., 2021b; Sun et al., 2023; Villegas et al., 2017; Oliu et al., 2018) on this task. Besides recurrently modeling, the sequence-to-sequence fashion (Gao et al., 2022; Tan et al., 2023) is employed with efficient U-Net structures to predict with a simplified configuration of convolutions. When minimizing mean error of predictions and uncertain future outcomes, these models usually generate blur appearance (Oprea et al., 2020). A straightforward solution is to exploit probabilistic models, like VAEs (Villegas et al., 2019; Wu et al., 2021a; Babaeizadeh et al., 2021), GANs (Clark et al., 2019; Tulyakov et al., 2018) and flow-based ones (Dorkenwald et al., 2021). However, they are highly likely to lead mode collapse and hard to fit (Oprea et al., 2020). As LDMs are dominant in generation-related works (Rombach et al., 2022; Ho et al., 2022b), we introduce LDMs in spatiotemporal sequence prediction for potential explorations in this field.

### 2.2 Latent Diffusion Models

Diffusion models (DMs) are one of likelihood-based generative models, revealing first remarkable results in image generation communities (Ho et al., 2020; Nichol et al., 2021) by progressively reversing a Markov chain which iteratively adds noise to target distributions (Ronneberger et al., 2015). Benefiting from lower computational requirement and better expressivity than DMs (Song et al., 2020; Karras et al., 2022), recently LDMs (He et al., 2022; Rombach et al., 2022) have made tremendous breakthroughs on various tasks, including T2I generation (Nichol et al., 2021; Rombach et al., 2022; Balaji et al., 2022; Saharia et al., 2022; OpenAI, 2023; Midjourney, 2023; Peebles & Xie, 2023; Podell et al., 2023), T2V generation (Shi et al., 2015; He et al., 2022; Yan et al., 2021; Singer et al., 2022; Voleti et al., 2022; Ho et al., 2022a; Blattmann et al., 2023b; Zhou et al., 2022; Wang et al., 2023; Midjourney, 2023), text-to-audio generation (Liu et al., 2023a), 3D shape generation (Vahdat et al., 2022), video editing (Liew et al., 2023), tabular data generation (Zhang et al., 2023a), video frame interpolation (Danier et al., 2024), etc. Most related directions are T2I and T2V models. In T2I generation (Nichol et al., 2021; Rombach et al., 2022; Balaji et al., 2022), novel images are generated with textual descriptions given as conditions, where representatives contain Dalle-2 (OpenAI, 2023), Midjourney (Midjourney, 2023), DiT (Peebles & Xie, 2023) and Stable Diffusion (Podell et al., 2023). T2V models are mostly inspired from T2I (Singer et al., 2022). VDM (Shi et al., 2015) reports first results by modifying 2D U-Net to a factorized 3D network to achieve video synthesis. Recent works include Imagen Video (Ho et al., 2022a), SORA (OpenAI, 2024), Make-A-Video (Singer et al., 2022), VideoGPT (Yan et al., 2021), MagicVideo (Zhou et al., 2022), Latte (Ma et al., 2024), StoryDiffusion (Zhou et al., 2024) and CogVideoX (Yang et al., 2024), extending existing image-based models to the video domain. They are usually built with two stages (Rombach et al., 2022). In the first stage, VQ-VAEs (Van Den Oord et al., 2017; Razavi et al., 2019) or VQ-GANs (Esser et al., 2021) are used as autoencoders to learn an expressive prior over discretized latent space. This autoencoder-diffusion paradigm has also been effectively adapted for sequential data synthesis, as demonstrated in TimeAutoDiff (Suh et al., 2024). In the second stage, a denoising network commonly implemented by U-Net (Ho et al., 2020; Ronneberger et al., 2015; Dhariwal & Nichol, 2021) is trained to predict the less noisy video samples progressively, reversing the diffusion process where Gaussian noise is iteratively added onto the raw data with predefined timesteps. Concurrently, efficient temporal modeling techniques like those in Latent-Shift (An et al., 2023) have advanced video generation by introducing specialized operations in the latent space. Beyond generation tasks, PriSTI (Liu et al., 2023b) addresses missing data challenges through conditional diffusion frameworks. Similarly, Stochastic Diffusion (Liu et al., 2024a) extends this capability to uncertainty-aware time series forecasting, highlighting the versatility of diffusion models in handling complex temporal dependencies. Different form T2I and T2V

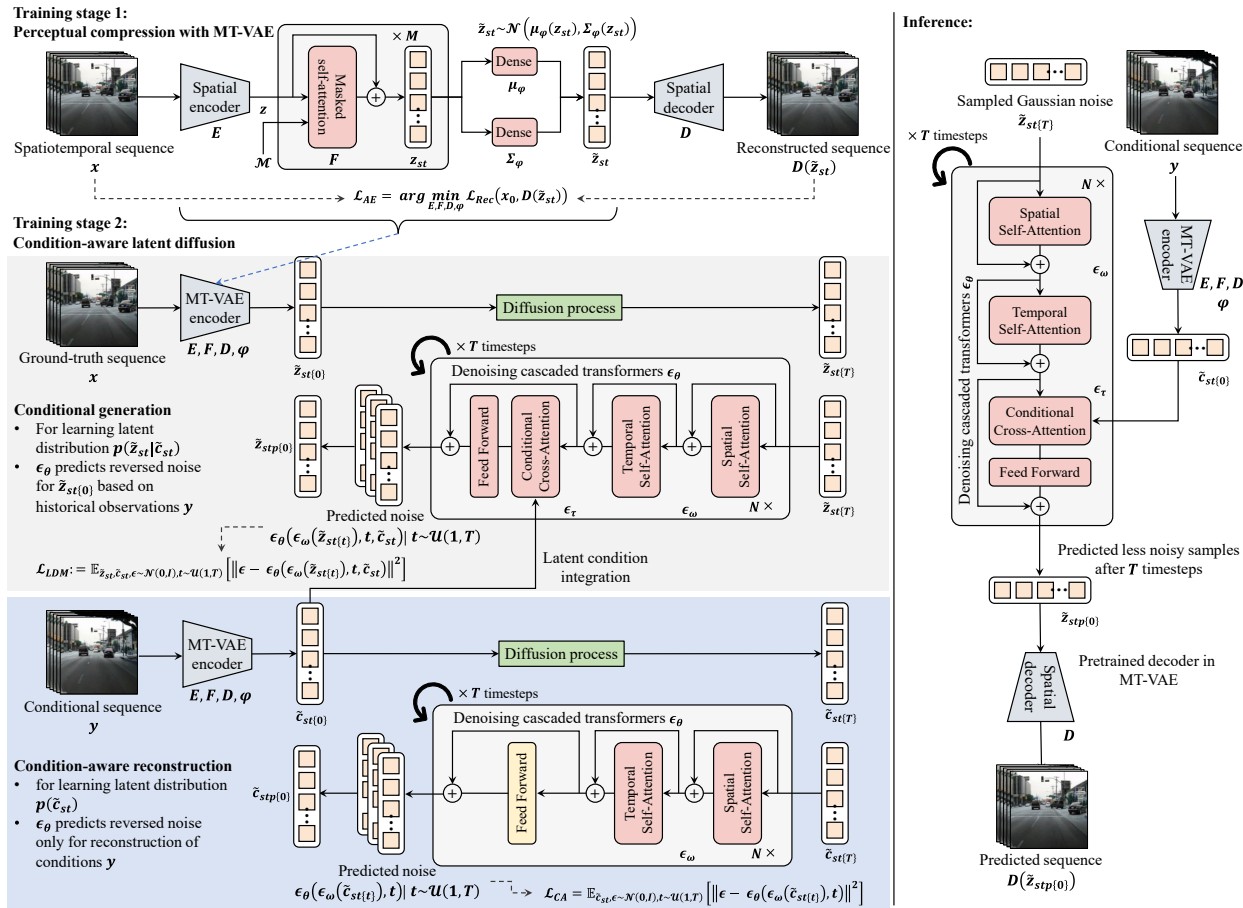

Figure 1: The pipeline of PredLDM. The training is comprised of two stages. In the first stage, MT-VAE is exploited to compress the spatiotemporal sequences into latent space. In the second stage, CA-LDM contains the learning of both conditional generation and condition-aware reconstruction. For inferencing, the Gaussian noise is sampled and less noisy latent vectors are predicted conditioned on the latent historical observations. The sampled latent vectors are then fed into the decoder for output.

models, the condition of our PredLDM is historical observations. This difference makes the condition is not as easy as text prompts used to be, as the spatiotemporal content is low dimensional and highly complex.

## 2.3 Compression of Spatiotemporal Sequences

In order to compress spatiotemporal data, 3D convolutions are straightforward solutions (Ho et al., 2022b; Tran et al., 2015). Given multiple frames, VDM (Ho et al., 2022b) and LVDM (He et al., 2022) exploit 3D U-Net convolutions by replacing each 2D convolutions in image models with space-only 3D convolutions. Instead of 3D convolutions expensive at fitting or hard to train, 1D convolutional temporal layers are attractive combined with 2D convolutions (Tran et al., 2018). Make-A-Video (Singer et al., 2022) initializes the spatial convolutional layers with pretrained T2I weights and adds temporal convolutions to correlate spatial features across time dimensions, similar as Imagen Video (Ho et al., 2022a), ModelScope (Wang et al., 2023), MagicVideo (Zhou et al., 2022) and Stable Video Diffusion (Blattmann et al., 2023a). There are also works combining convolutional temporal layers with 3D U-Net (Blattmann et al., 2023b). Although convolutions are effective in image modeling, diverse and global relations in time dimensions are too complex limited for their local receptive field (Li et al., 2023). To additionally capture intricate global dependencies, we make attempts by proposing MT-VAE.

## 3 Methods

The training of PredLDM includes two stages. In the first stage, PredLDM learns to compress spatiotemporal sequences into latent vectors by MT-VAE to model intricate global coherence. In the second stage, CA-LDM is designed for comprehensive understanding of historical spatiotemporal content in conditions. To inference, latent noise is randomly sampled and denoised with trained LDMs conditioned on historical embeddings. Predicted latent vectors are finally fed to the decoder for future content.

### 3.1 Background on Latent Diffusion Models

**Diffusion formulation.** Denoising diffusion probabilistic models (Ho et al., 2020) simulate a data distribution $x \sim p_{data}(x)$ by corrupting data with progressively added Gaussian noise and learning to reverse this process. The diffusion process leads corrupted data resembling pure noise by gradually adding Gaussian noise in a serious of timesteps, along with the sampled noisy $x_t$ at timestep $t$,

$$q(x_t|x_{t-1}) = \mathcal{N}\big(x_t; \sqrt{1-\beta_t}x_{t-1}, \beta_t I\big), \tag{1}$$

where $\{\beta_t\}_{t=1}^T$ are a set of linearly increasing hyperparameters with the predefined variance schedule (He et al., 2022), $T$ denotes the number of diffusion steps and $\mathcal{N}$ refers to the normal distribution. The denoising process reverses the above diffusion process to predict less noisy $x_{t-1}$ iteratively,

$$p_\theta(x_{t-1}|x_t) = \mathcal{N}\big(x_{t-1}; \mu_\theta(x_t, t), \Sigma_\theta(x_t, t)\big), \tag{2}$$

where $\mu_\theta$ and $\Sigma_\theta$ are accomplished by a parameterized denoising model $\epsilon_\theta$ with learnable parameters $\theta$. Specifically, $\epsilon_\theta(x_t, t)$ is trained to predict the noise at each step of the diffusion process by minimizing the difference between the actual noise and predicted ones,

$$\mathcal{L}_{DM} = \mathbb{E}_{x_0, \epsilon \sim \mathcal{N}(\mathbf{0}, \mathbf{I}), t \sim \mathcal{U}(1, T)}[\|\epsilon - \epsilon_\theta(x_t, t)\|^2], \tag{3}$$

where $\mathcal{U}(1, T)$ refers uniformly sampling from $\{1, \cdots, T\}$.

**Latent diffusion models.** LDMs (Rombach et al., 2022) are efficient variants of DMs by operating in latent space. This process begins with a pretrained variational encoder $E: x \to z$, compressing the input image $x \sim p_{data}(x)$ into latent representations $z \sim E(x)$. Similar as Equation 1 and Equation 2 with $z$,

$$\mathcal{L}_{LDM} = \mathbb{E}_{E(x_0), \epsilon \sim \mathcal{N}(\mathbf{0}, \mathbf{I}), t \sim \mathcal{U}(1, T)}[\|\epsilon - \epsilon_\theta(z_t, t)\|^2]. \tag{4}$$

### 3.2 Perceptual Compression with MT-VAE

In the first training stage, we compress spatiotemporal sequences with MT-VAE. The model structure of MT-VAE is provided as Figure 1. Given a sequence $x_0 \sim p_{data}(x_0)$, $x_0 \in \mathbb{R}^{L \times H \times W \times C}$, where $L$, $H$, $W$ and $C$ are the temporal length, height, width and channel number respectively, the spatial convolutional encoder $E$ encodes $x_0$ into latent vectors $z_0 = E(x_0)$. $z_0$ is taken by masked temporal self-attention modules $F$, extracting intricate coherent temporal reliance,

$$z_{st} = \mathrm{mAtt}\left(Norm(z_s)\right) + z_s, z_s = (E(x_0) + U), \tag{5}$$

$$z_{st} := \mathrm{FeedForward}\big(Norm(z_{st})\big) + z_{st}, \tag{6}$$

where $U$ is the positional embedding obtained by convolutions of input. Insipred by scaled dot-product attention (Vaswani, 2017) and masked modeling (Xie et al., 2022; Cheng et al., 2022), mAtt$(\cdot)$ is to capture complex global reliance after the layer normalization. Multi-head mechanism (Vaswani, 2017) is used to project representations into subspaces calculated by different attention heads, the number of heads is denoted by $H$. The process of mAtt$(\cdot)$ is defined as below, assuming that normalized features of $z_s$, $Norm(z_s)$ is $z_{in}$,

$$\mathrm{mAtt}(z_{in}) = \mathrm{mAtt}(Q, K, V, \mathcal{M}), \tag{7}$$

$$\left(Q^{(i)}, K^{(i)}, V^{(i)}\right) = z_{in}\left(W^{(Q,i)}, W^{(K,i)}, W^{(V,i)}\right) \tag{8}$$

$$z_{out}^{(i)} = \text{softmax} \left( \frac{Q^{(i)} K^{(i)T}}{\sqrt{d_k}} + \mathcal{M} \right) V^{(i)}, \tag{9}$$

$$\text{where} \quad \mathcal{M}_j = \begin{cases} 0, & \text{if} \quad md(j) = 1, \\ -\infty, & \text{otherwise}, \end{cases} \tag{10}$$

$$\text{mAtt}(Q, K, V) = \text{Concat} \left( z_{out}^{(1)}, \cdots, z_{out}^{(H)} \right) W^O, \tag{11}$$

where $Q$, $K$ and $V$ are queries, keys and values of vectors for dot production (Vaswani, 2017). $W^{(Q,i)}$, $W^{(K,i)}$ and $W^{(V,i)}$ are the parameters of linear operations for $i$-th head to control the weights of $Q$, $K$ and $V$ respectively, $i = 1, \cdots, H$. Here, the masking is operated on the attention matrix via Equation 9 by $\mathcal{M}$, where we can see this masking modulating the scaled dot production between queries and keys, $md(j) \in \{0, 1\}^L$ is the random binarized output with the masking ratio $r$ of the same size as temporal length. $\mathcal{M}_j$ indicates all zero or all negative infinite matrices corresponding to time location $j$. The results from different attention heads are concatenated and projected back into representation space through the weight matrix $W^O$. Here the compressed feature $z_{st}$ is accessed. The mean vectors $\mu_\varphi(z_{st})$ and variance vectors $\Sigma_\varphi(z_{st})$ are predicted with learnable parameters $\varphi$, which are implemented by two dense layers. The sampled latent features from the Gaussian distribution $\tilde{z}_{st} \sim \mathcal{N}(\mu_\varphi(z_{st}), \Sigma_\varphi(z_{st}))$ are the final compressed features used for the decoder $D$ to reconstruct $x_0$. $D$ is accomplished by the cascaded convolutional layers and $D(\tilde{z}_{st})$ is expected to minimize the difference between the predicted distributions and the real data $p_{data}(x_0)$. The training objective is the reconstruction loss with a pixel-level mean-squared error (MSE) and a perceptual loss (Johnson et al., 2016; Ni et al., 2023). For compression, varying spatiotemporal patterns can be accessed by our MT-VAE, as the masking operation creates more attention matrix-based samples along time dimensions. The attention matrix in our work is accessed from dense convolutions from previous latent input samples. By creating more intermediate attention feature samples, complex fractions of the same piece of spatiotemporal sequences can be linked tightly.

### 3.3 Condition-aware Latent Diffusion

In the second stage, it is expected to train a denoising network to predict less noisy samples from latent noise conditioned on historical embeddings (Sohn et al., 2015; Rombach et al., 2022). Different from existing video generation research, the condition of this task is raw historical spatiotemporal sequences rather than high dimensional text as conditions, while spatiotemporal sequences are full of structural details, more difficult to comprehend. Our diffusion part consists of both the commonly used conditional generation objection and importantly the probabilistic modeling of the condition itself, maximizing the mutual information beyond joint distributions of conditional generation only. Given a sequence of future observations $x_0 \sim p_{data}(x_0)$ and its corresponding historical observations as conditions $y_0 \sim p_{data}(y_0)$, the trained MT-VAE compresses them into the latent features respectively as $\tilde{z}_{st}$ and $\tilde{c}_{st}$, $\tilde{z}_{st} \sim \mathcal{N}(\mu_\varphi(z_{st}), \Sigma_\varphi(z_{st}))$, $z_{st} = F(E(x_0))$ and $\tilde{c}_{st} \sim \mathcal{N}(\mu_\varphi(c_{st}), \Sigma_\varphi(c_{st}))$, $c_{st} = F(E(y_0))$. Our CA-LDM is trained to simultaneously learn the denoising diffusion process of (i) conditional generation on the distributions $p(\tilde{z}_{st}|\tilde{c}_{st})$ and (ii) reconstruction of the conditions on the distribution $p(\tilde{c}_{st})$, as in Figure 1. For pseudo-code, please refer to Algorithm 1.

**Conditional generation.** For learning $p(\tilde{z}_{st}|\tilde{c}_{st})$, the diffusion process progressively adds Gaussian noise onto $\tilde{z}_{st}$ until it resembles pure noise along with the sampled noisy $\tilde{z}_{st\{t\}}$ at timestep $t$. The denoising process reverses the diffusion process iteratively to approach the original latent samples $\tilde{z}_{st\{0\}}$. Instead of using time-conditional U-Net (Ronneberger et al., 2015), our denoising network $\epsilon_\theta$ is inspired by DiTs (Ma et al., 2024). It is consisted of the spatiotemporal self-attentions $\epsilon_\omega$ and cross-attentions $\epsilon_\tau$, where $\omega$ and $\tau$ are learnable parameters. This denoising neural network is trained by minimizing the difference between the actual noise and predicted ones,

$$\mathcal{L}_{LDM} := \mathbb{E}_{\tilde{z}_{st}, \tilde{c}_{st}, \epsilon, t} \left[ \left\| \epsilon - \epsilon_\theta(\epsilon_\omega(\tilde{z}_{st\{t\}}), t, \tilde{c}_{st}) \right\|^2 \right], \tag{12}$$

where $t \sim \mathcal{U}(1, T)$ and $\epsilon \sim \mathcal{N}(\mathbf{0}, \mathbf{I})$. The condition latent features $\tilde{c}_{st}$ is integrated to the intermediate features of $\epsilon_\theta$ by the cross-attention layers $\epsilon_\tau$.

**Condition-aware reconstruction.** For learning $p(\tilde{c}_{st})$, the diffusion process is additionally conducted onto $\tilde{c}_{st}$ with the sampled noisy $\tilde{c}_{st\{t\}}$ at timestep $t$. The denoising phase is operated to reverse the noisy samples to the original distribution $p(\tilde{c}_{st})$. The training objective of this branch is to progressively reconstruct the conditions by minimizing the difference between the actual noise and predicted ones from the denoising network with Siamese spatiotemporal self-attentions $\epsilon_\omega$ in $\epsilon_\theta$ yet without cross-attentions,

$$\mathcal{L}_{CA} = \mathbb{E}_{\tilde{c}_{st},\epsilon,t}\left[\left\|\epsilon - \epsilon_\theta(\epsilon_\omega(\tilde{c}_{st\{t\}}),t)\right\|^2\right]. \tag{13}$$

The overall learning objective of CA-LDM is the combination of $\mathcal{L}_{LDM}$ and $\mathcal{L}_{CA}$,

$$\mathcal{L}_{CA-LDM} = \mathcal{L}_{LDM} + \mathcal{L}_{CA}. \tag{14}$$

---

**Algorithm 1** Condition-aware Latent Diffusion (CA-LDM) Training

---

1: **Input:**
2:     Future observations: $x_0 \sim p_{data}(x_0)$; historical observations (condition): $y_0 \sim p_{data}(y_0)$;
3: **Pretrained network:**
4:     Pre-trained MT-VAE encoder $E$, transformer $F$, and parameter $\varphi$;
5: **Hyperparameters:**
6:     Number of diffusion steps $T$; variance schedule $\{\beta_t\}_{t=1}^T$;
7: **Initialization:**
8:     Initialized denoising network $\epsilon_\theta$ with parameters $\theta = \{\omega, \tau\}$;
9: **procedure** CA-LDM Training
10:     Compress observations into latent space:
11:     $z_{st} = F(E(x_0))$; $c_{st} = F(E(y_0))$;
12:     Sample latent features:
13:     $\tilde{z}_{st} \sim \mathcal{N}(\mu_\varphi(z_{st}), \Sigma_\varphi(z_{st}))$; $\tilde{c}_{st} \sim \mathcal{N}(\mu_\varphi(c_{st}), \Sigma_\varphi(c_{st}))$;
14:     **for** $t = 1$ **to** $T$ **do**
15:        Sample random noise: $\epsilon \sim \mathcal{N}(0, \mathbf{I})$; $t \sim \mathcal{U}(1, T)$;
16:        Add noise to latent features:
17:        $\tilde{z}_{st(t)} = \sqrt{\bar{\alpha}_t}\tilde{z}_{st} + \sqrt{1 - \bar{\alpha}_t}\epsilon$; $\tilde{c}_{st(t)} = \sqrt{\bar{\alpha}_t}\tilde{c}_{st} + \sqrt{1 - \bar{\alpha}_t}\epsilon$; where $\bar{\alpha}_t = \prod_{s=1}^t (1 - \beta_s)$;
18:        Compute conditional generation loss:
19:        $\mathcal{L}_{\text{LDM}} = \mathbb{E}\left[\|\epsilon - \epsilon_\theta(\epsilon_\omega(\tilde{z}_{st(t)}), t, \tilde{c}_{st})\|^2\right]$
20:        Compute condition-aware reconstruction loss:
21:        $\mathcal{L}_{\text{CA}} = \mathbb{E}\left[\|\epsilon - \epsilon_\theta(\epsilon_\omega(\tilde{c}_{st(t)}), t)\|^2\right]$
22:        Update parameters:
23:        $\theta \leftarrow \theta - \nabla_\theta(\mathcal{L}_{\text{LDM}} + \mathcal{L}_{\text{CA}})$
24:     **end for**
25: **end procedure**
26: **Output:** Trained denoising network $\epsilon_\theta$

---

## 3.4 Inference

As in Figure 1, to inference the expected spatiotemporal sequence $x_0$ with the condition $y_0$, a Gaussian noise $z_T$ is sampled in the latent space and the condition is compressed by MT-VAE as $\tilde{c}_{st}$. The denoising network $\epsilon_\theta$ in CA-LDM is used to predict the less noisy samples $z_{0:T-1}$ within the predefined $T$ timesteps, while the latent condition vectors $\tilde{c}_{st}$ are fused across cross-attention layers into the intermediate features of $\epsilon_\theta$ to accomplish conditional controlling. The less noisy sample can be predicted by $z_{t-1} \sim \mathcal{N}(z_{t-1}; \mu_\theta(z_t, t), \Sigma_\theta(z_t, t))$. The latent vectors $z_0$ can be approached after $T$ timesteps denoising. Finally, the decoder in MT-VAE decodes $z_0$ back to the pixel space, resulting predictions as close as possible to ground-truth $x_0$.

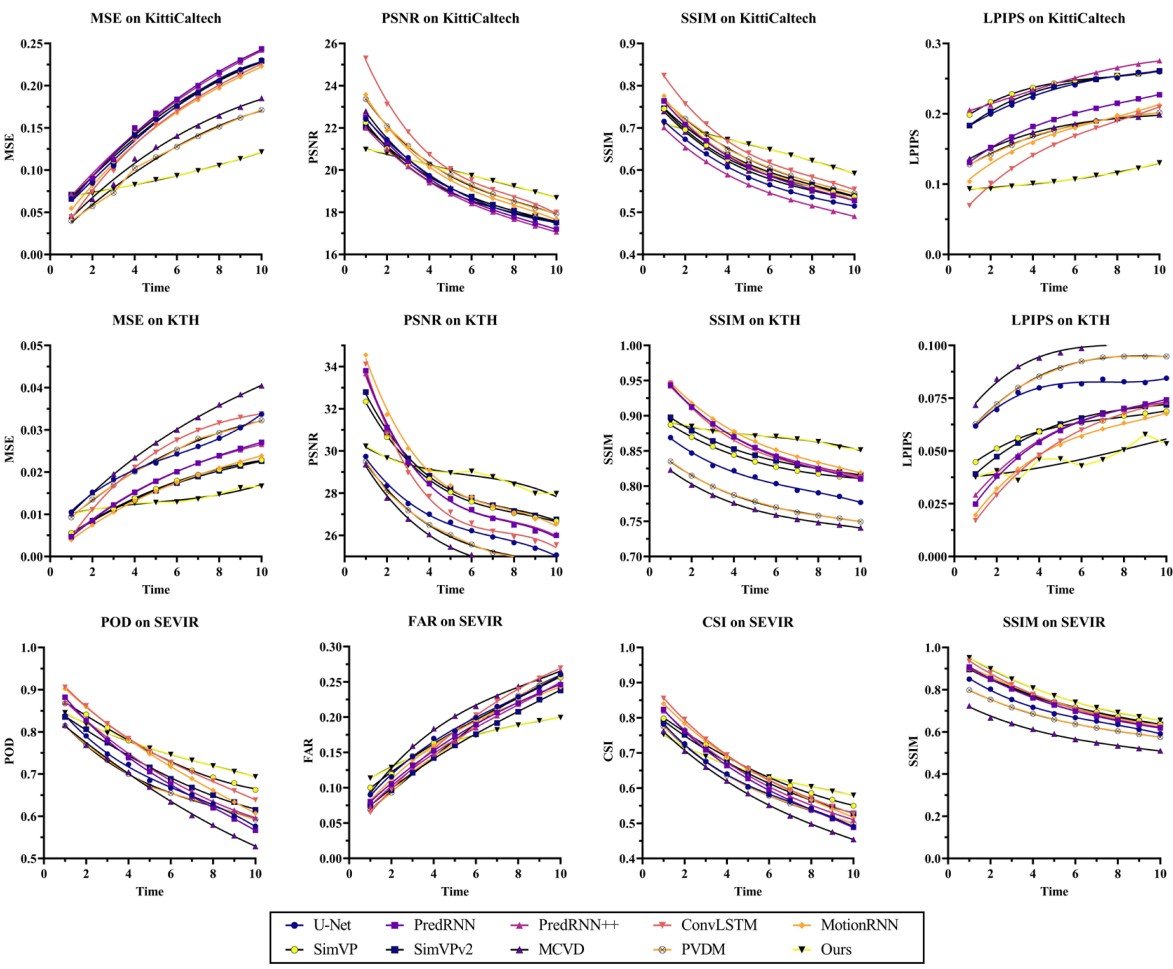

Figure 2: Temporal analysis of predictive models. Rows are from KittiCaltech, KTH and SEVIR respectively.

# 4 Experiments

## 4.1 Experimental Setup

**Datasets and metrics.** Datasets in this study include KittiCaltech (Geiger et al., 2013), KTH (Schuldt et al., 2004) and SEVIR (Veillette et al., 2020). (i) KittiCaltech is a driving-scene dataset, comprising a curated collection of high-quality images. The ability to predict the future dynamics of this scenario is paramount for the advancement of autonomous driving technology and dynamic comprehension, containing $127,271$ frames in total, with $74,833$ frames for training and $52,438$ frames for testing. (ii) KTH stands as a benchmark in the field of human action recognition and prediction. It encompasses diverse image sequences depicting a variety of human activities. This dataset is comprised of $51,360$ frames with $20,420$ frames for training and $30,940$ frames for testing. (iii) SEVIR has been curated in the realms of weather sensing and short-term forecasting, comprising thousands of weather events in multipe sensor modalities. We use vertically integrated liquid (VIL) data with a 5-minute interval, and $1\ km$ spatial resolutions. They are stored as integers ranging from 0 to 254, with a value of 255 indicating missing data. In our experiments, all frames are processed as $128 \times 128$ resolutions. The temporal length of input is uniformly 10 frames and output length is also 10 frames. For evaluating on KittiCaltech and KTH, the metrics contain MSE, PSNR, SSIM (Jin et al., 2020; Wang et al., 2004) and LPIPS (Zhang et al., 2018). For evaluating weather patterns on SEVIR, we use event-level short-term prediction metrics and the image quality assessment metric, including

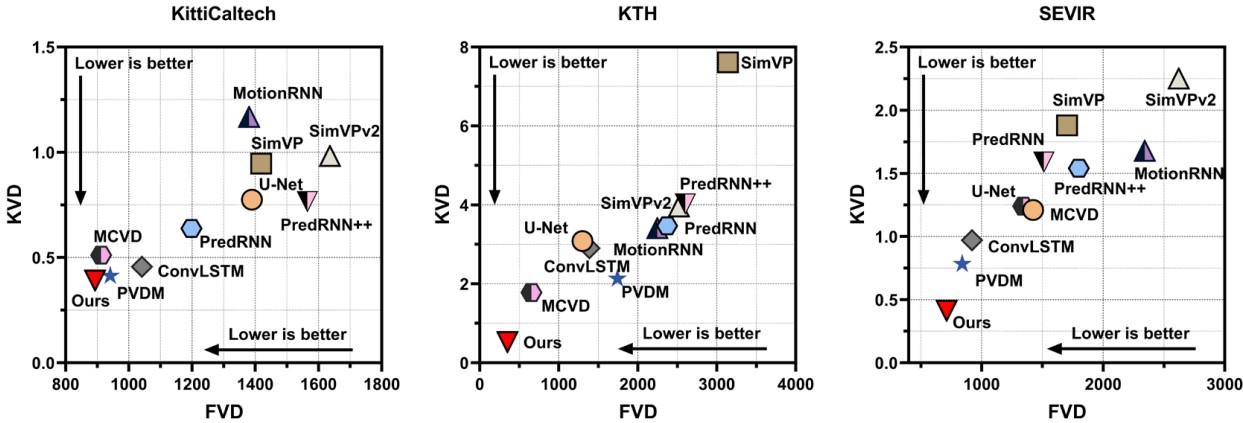

Figure 3: Plots of KVD/FVD distance scores from existing models and ours on three datasets. Distance scores between distributions of ground-truth data and predictions from predictive models are visualized. Both metrics are the lower the better.

Table 1: Comparison between existing models and PredLDM on KittiCaltech, KTH as well as SEVIR datasets. ↑ indicates the higher the better, whereas ↓ is the opposite. The best results are marked as bold and the second best ones are marked as underline.

| Models | KittiCaltech | | | KTH | | | SEVIR | | |
|---|---|---|---|---|---|---|---|---|---|
| | PSNR ↑ | SSIM ↑ | LPIPS ↓ | PSNR ↑ | SSIM ↑ | LPIPS ↓ | POD ↑ | FAR ↓ | CSI ↑ |
| U-Net | 19.34 | 0.591 | 0.232 | 26.75 | 0.813 | 0.079 | 0.691 | 0.185 | 0.612 |
| PredRNN | 19.19 | 0.616 | 0.190 | 28.33 | 0.860 | 0.057 | 0.704 | 0.174 | 0.629 |
| PredRNN++ | 18.99 | 0.571 | 0.244 | 28.33 | 0.861 | 0.058 | 0.714 | 0.171 | 0.638 |
| ConvLSTM | **20.46** | 0.652 | 0.154 | 27.87 | 0.859 | 0.052 | 0.754 | 0.182 | 0.660 |
| MotionRNN | 19.79 | 0.621 | 0.170 | 28.94 | 0.868 | 0.051 | 0.742 | 0.177 | 0.654 |
| SimVP | 19.19 | 0.614 | 0.239 | 28.47 | 0.838 | 0.060 | 0.752 | 0.184 | 0.657 |
| SimVPv2 | 19.29 | 0.620 | 0.234 | 28.64 | 0.847 | 0.061 | 0.713 | 0.163 | 0.641 |
| MCVD | 19.41 | 0.607 | 0.177 | 25.78 | 0.770 | 0.094 | 0.659 | 0.197 | 0.583 |
| PVDM | 19.94 | 0.631 | 0.174 | 26.25 | 0.781 | 0.086 | 0.681 | 0.173 | 0.610 |
| PredLDM | 19.86 | **0.653** | **0.107** | **28.94** | **0.871** | **0.045** | **0.760** | **0.148** | **0.672** |

POD (Veillette et al., 2020), FAR (Veillette et al., 2020), CSI (Schaefer, 1990) and SSIM. More details can be accessed as in Supplementary Section A.1.

**Implementation details.** PredLDM is trained in two stages. In the first stage, spatiotemporal sequences are autoencoded by MT-VAE. The masking ratio $r$ is set as 0.6. In the second stage, trained parameters in MT-VAE including $E$, $F$ and $\varphi$ are used to project data into latent space. The denoising model is trained to predict less noisy samples by the linear combination of $\mathcal{L}_{LDM}$ and $\mathcal{L}_{CA}$. More details and hyper parameters implementing our PredLDM are available as in Supplementary Section A.1.

## 4.2   Results with Comparison to Existing Models

Baselines in our experiments include classical encoder-forecaster architectures U-Net (Ronneberger et al., 2015), PredRNN (Wang et al., 2017), PredRNN++ (Wang et al., 2018), ConvLSTM (Shi et al., 2015), MotionRNN (Wu et al., 2021b), SimVP (Gao et al., 2022), SimVPv2 (Tan et al., 2023) as well as diffusion-based probabilistic generation architectures MCVD (Voleti et al., 2022) and PVDM (Yu et al., 2023). As in Table 1, PredLDM almost achieves best scores in all metrics on KittiCaltech and KTH, except in PSNR. The most shining metric is LPIPS which resembles the perceptual evaluating abilities similar to human,

Table 2: Influence of different settings of autoencoders. ↑ indicates the higher the better, whereas ↓ is the opposite. The best results are marked as bold.

| Autoencoders | KittiCaltech | | KTH | |
|---|---|---|---|---|
| | SSIM ↑ | LPIPS ↓ | SSIM ↑ | LPIPS ↓ |
| 3D VAEs | 0.630 | 0.157 | 0.861 | 0.060 |
| 2D VAEs + 1D Convs | 0.647 | 0.121 | 0.869 | 0.045 |
| MT-VAE (Ours): | **0.653** | **0.107** | **0.871** | **0.045** |

Table 3: Influence of condition-aware latent diffusion. ↑ indicates the higher the better, whereas ↓ is the opposite. The best results are marked as bold.

| LDMs | KittiCaltech | | KTH | |
|---|---|---|---|---|
| | SSIM ↑ | LPIPS ↓ | SSIM ↑ | LPIPS ↓ |
| LDM $\mathcal{L}_{LDM}$ | 0.622 | 0.184 | 0.858 | 0.067 |
| CA-LDM (Ours): LDM $\mathcal{L}_{LDM} + \mathcal{L}_{CA}$ | **0.653** | **0.107** | **0.871** | **0.045** |

where 30.5% improvement is achieved by PredLDM on KittiCaltech and 11.8% improvement is made on KTH. Results on weather nowcasting show the large margin in scores of PredLDM over other models.

### 4.3 Temporal Analysis

For observing temporal detailed performance, results are compiled by performance at each time point as in Figure 2. It is evident that the descending curve decreases slowly for PredLDM, while the degradation of other models is quite fast. This implies that PredLDM is capable of dealing with complex global temporal variations. This phenomenon is more prominent in LPIPS on KittiCaltech and KTH.

### 4.4 Experiments on KVD/FVD Comparison

For further analyzing the temporal retention ability of models, we calculate Fréchet Video Distance (FVD) and Kernel Video Distance (KVD) (Unterthiner et al., 2018) between predictions and groundtruth on KittiCaltech, KTH and SEVIR datasets. Different from frame-level metrics, these two metrics measure the similarity of sequential distributions. As in Figure 3, results indicate better retention abilities of visual quality in temporal dimensions.

### 4.5 Predictive Error Analysis

For analyzing the predictive error between existing competitive implementations and ours, we select a challenging sample when a car is driving through the corner. As in Figure 4, the visual appearance from PredLDM shares quite similar details as the ground truth while others show evident visual difference. For visualizing the accumulated error over time on these models, we count the cumulative values of all absolute errors along the x-axis direction through time. The plot shows that the accumulated error over time is smaller for predictions from PredLDM than others. For the error map calculated from the last predicted frame, the error of the results is very limited from ours. It can be observed that the visual quality and accumulative error are evidently improved by our model.

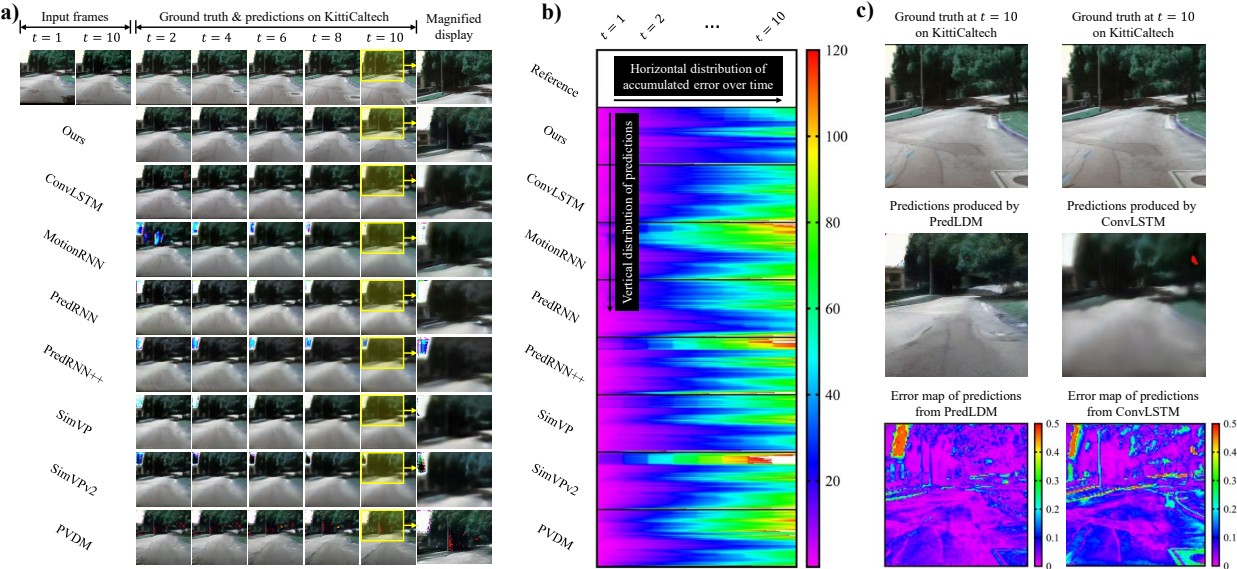

Figure 4: Predictive error analysis. **a),** A challenging case on KittiClatech is visualized. **b),** Analysis on error of predictions accumulated with time. We calculate the accumulated error along with x-axis, the distribution should be all blue if there is no error accumulated. **c),** Error maps of the last frame. The highlighted value indicates the largest error.

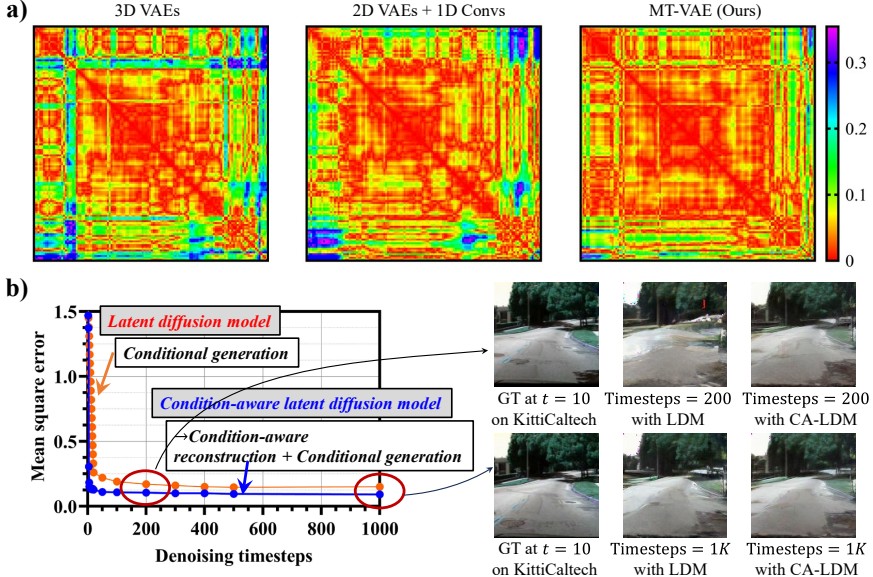

Figure 5: Ablation study. **a),** Heatmaps of correlation between synthetic data v.s. real data for MT-VAE. **b),** Influence of condition-aware latent diffusion. Performance in denoising processes is plotted on the left and the decoded predictions are on the right.

## 4.6 Ablation Study

For organizing the ablation study, we firstly compile the performance on different settings of autoencoders including 3D VAEs, 2D VAEs + 1D convolutions and ours MT-VAE, as in Table 2. It can be seen that the setting of 2D VAEs with 1D convolutional attention is more competitive than 3D VAEs, consistent with existing research (Tran et al., 2018). Our setting of MT-VAE is better than this competitive one. Then

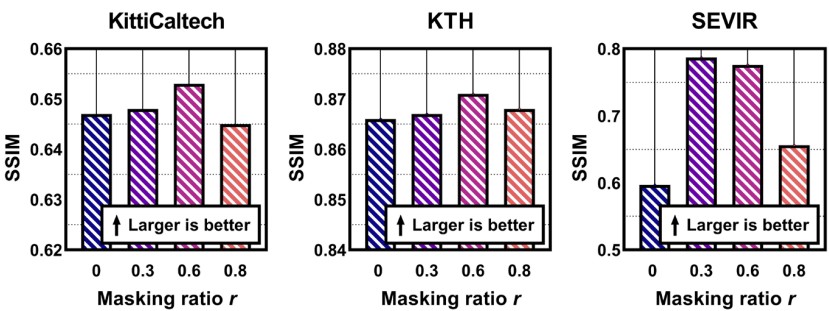

Figure 6: Influence of masking ratios on forecasting performance. SSIM scores with different masking ratios of PredLDM on KittiCaltech, KTH and SEVIR are reported.

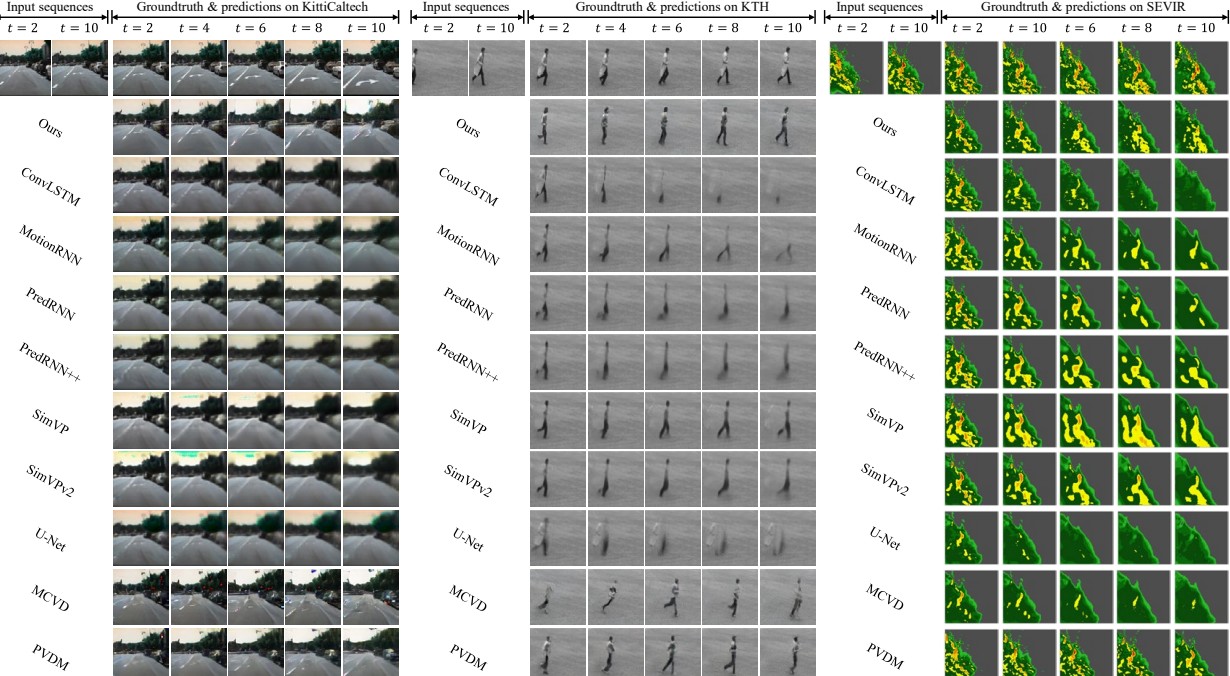

Figure 7: Challenging cases on three datasets. Visualized challenging cases on KittiCaltech, KTH and SEVIR are presented. The first row refers to input and ground-truth spatiotemporal sequences, following rows indicate predictions.

we report the influence of condition-aware latent diffusion as in Table 3, showing that the condition-aware reconstruction-based loss is beneficial.

Further study is conducted as Figure 5. Correlations are analyzed from settings of autoencoders in Table 2. It shows that our MT-VAE is better for generating realistic data, where the correlation-based distributions of ours indicate the most similar behavior as realistic structures. The mean square error of sampled results during the denoising process shows that the addition of condition-aware constraint brings lower error. From the decoded visual appearance, CA-LDM can produce more realistic visual quality. More analysis is available in Supplementary Section A.2.

### 4.7 Influence on Masking Ratios

For investigating the influence of the masking ratio $r$ on forecasting performance on three datasets, different settings of masking ratios are experimented as in Figure 6. More results are provided as in Supplementary Section A.3. Firstly, when the masking ratio is 0, the setting refers to the masking mechanism does not work in MT-VAE and instead the non-masked transformer-based VAE compresses spatiotemporal data. Comparing against the non-masked settings, it can be witnessed that the masked attention in MT-VAE is effective in most aspects on three datasets. Meanwhile, these results also indicate that the setting of masking ratio is preferred as 0.6 in our experiments.

### 4.8 Case Study

Predictions of existing models and ours PredLDM on challenging cases of three datasets can be accessed in Figure 7. It can be seen that the predictions form PredLDM not only show the realistic visual appearance, but also share the most similar movement as the ground-truth sequences. More examples are available in Supplementary Section A.10.

### 4.9 Discussion and Limitation

Distortion-perception trade-off. We have observed a phenomenon from our experimental results especially in Table 1 that some classical predictive models such as ConvLSTM and MotionRNN behave excellent performance in structured-based metrics PSNR and SSIM, while not good at perceptual metric LPIPS. This states the trade-off in this context. Distortion (measured by PSNR and SSIM) and perceptual quality (LPIPS) are at odds with each other. Through our experiments, it can be found out that classical deterministic models excel in PSNR/SSIM because they are optimized for pixel-level fidelity, while probabilistic ones excel in perceptual metrics because they are optimized for statistical realism and diversity of outputs. This reveals the selection preference for specific downstream usage. For example, in some accuracy required occasions for example scientific measurement, the easy-implement and accurate classical models are preferred, while for some realistic necessary usages, probabilistic modeling may be an excellent choice.

Time-evolving performance. For autoregressive models like ConvLSTM, PredRNN and MotionRNN, a process generating frames or tokens one at a time. For sequence-to-sequence models like U-Net and SimVP, the whole spatiotemporal sequence is predicted by deterministic optimization. For diffusion-based models, the denoising process runs for a predefined number of steps to predict results from noise. From results in Figure 2, Figure 4 and Figure 5, it can be observed that the diffusion-based modeling is beneficial to reduce time iterative error. Besides, the proposed PredLDM can serve as a competitive baseline in this field.

Masked operations in attention matrix are beneficial to temporal modeling. DiT (Ma et al., 2024) proposes two kinds of masking strategies, including simple masking from one-hot notes on time tokens and frame-level masking, which is operated on the direct pixel of frames. SVD (Blattmann et al., 2023a) additionally exploits a binary mask where 1 indicates the presence of a conditioning frame and 0 that of a mask embedding on the input of U-Net denoising network. We can see their masking is on the input of denoising network by concatenating a masking conditioning frame-size embedding. Make-A-Video (Singer et al., 2022) uses the image-based masking trained for frame interpolation, still limited on data-level. MCVD (Voleti et al., 2022) masks frames in two ways on both past and future frames. Different from these methods, our MT-VAE is operated on the attention matrix, especially on the scaled dot production matrix from query and key in transformers of VAE. From results in Figure 6 and Figure 10, it can be observed that our masked modeling is also beneficial against the version without masking (when the masking ratio is 0). For future research, more deep research on attention matrix-based masked modeling may be of much value.

Real-time performance. While our model demonstrates competitive performance, it is important to acknowledge its computational limitations. Although the model scale and computational cost are not excessive compared to the diffusion baseline (MCVD) and our inference speed of 1.33 seconds per sequence is faster, the current setting is not ideal yet for real-world fast applications satisfying real-time usage such as autonomous driving. Specifically, generating 10 frames takes 1.331 seconds, even with the compression provided by the

MT-VAE. Therefore, for practical real-time deployment, further acceleration techniques would be necessary. For offline usages like urban supervision, analyzing or weather forecasting, it may remain capable of serving.

## 5 Conclusion

In this study, we propose a spatiotemporal predictive model with LDMs called PredLDM, towards predicting the accurate and realistic future. Under consideraton of intricate global coherence and comprehensive history understanding, corresponding designs are made. (i) MT-VAE is proposed to compress intricate global coherent spatiotemporal latent representations with the combination of transformers with masked attention and convolutional VAEs. (ii) CA-LDM is proposed by learning distributions of both conditional generation and condition-aware reconstruction, to comprehensively understand conditions which are spatiotemporal sequences with diverse and complex context. Through extensive experiments on multiple scenarios, PredLDM shows accurate performance and realistic appearance in predictions, revealing promising potential in future research and applications.

### Acknowledgments

Being supported by The National Natural Science Foundation of China (Nos. 42075139, 42077232, 42075139, 82404639); The Science and technology program of Jiangsu Province (No. BE2020082, BE2022063); Natural Science Foundation of Jiangsu Province (BK20241249); National Model 'Digital Intangible Cultural Heritage' Immersive Training Base for VET.

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

# A Appendix

## A.1 More Details on Data and Implementation

**Data.** Datasets in this study include driving scene-related dataset KittiCaltech (Geiger et al., 2013), human action-related dataset KTH (Schuldt et al., 2004) and weather pattern-related dataset SEVIR (Veillette et al., 2020). (i) KittiCaltech dataset is a cornerstone in the domain of computer vision, serving as an essential resource for autonomous driving research. It comprises a curated collection of high-quality images that are vital for the understanding of driving scenarios. The ability to predict the future dynamics of these scenarios is paramount for the advancement of autonomous driving technology, rendering this dataset

exceptionally valuable for research in vision field, particularly in the areas of future scenario prediction and dynamic comprehension. This dataset is meticulously organized, consisting of a total of $127,271$ frames. Within this collection, $74,833$ frames are allocated for training purposes, while $52,438$ frames are reserved for testing. This structured distribution ensures a comprehensive framework for both the development and validation of autonomous driving algorithms. (ii) KTH dataset stands as a benchmark in the field of human action recognition and prediction. It encompasses a diverse array of image sequences depicting a variety of human activities, such as walking, jogging, running, boxing, waving, and clapping, totaling six distinct categories, capturing the intricacies of different individuals performing various actions. Comprising a total of $51,360$ frames, the KTH dataset is segmented into $20,420$ frames for training and $30,940$ frames for testing. To maintain uniformity in evaluation, all frames are centrally cropped and resized to a consistent dimension of $128 \times 128$ pixels. The dataset's processing protocol specifies that the input consists of 10 frames, with the output also comprising 10 frames, ensuring a standardized framework for analysis and comparison. (iii) SEVIR dataset has been curated to accelerate research in the realms of weather sensing, avoidance and short-term forecasting. This comprehensive collection comprises thousands of weather events with each represented as a 4-hour sequence. Researchers are empowered to synthesize and harmonize diverse weather sensor data into a unified dataset through SEVIR. The dataset encompasses a variety of sensor modalities, including IR069 (infrared satellite imagery at 6.9 m), IR107 (infrared satellite imagery at 10.7 m), VIL (vertically integrated liquid), and LGHT (Lightning). In this study, we use VIL modality. The VIL data is derived from NEXRAD radar mosaics, featuring a $384 \times 384$ pixel resolution, a 5-minute interval, and a 1 km spatial resolution. The geographically and chronologically aligned imagery, depicting a spectrum of weather events including high winds, tornadoes, and hail, is captured by GEOS-16 satellites and NEXRAD weather radars. This data is publicly available in HDF files, we convert them into recordings of images. The pixel values within these spatial grids correspond to processed statistics derived from the actual sensor readings. VIL images are stored as integers ranging from 0 to 254, with a value of 255 indicating missing data. All frames have been reprocessed to a $128 \times 128$ resolution. The temporal length of input is uniformly 10 frames and output length is also 10 frames.

**Implementation Details.** The training of PredLDM is comprised of two stages. In the first stage, spatiotemporal sequences are autoencoded by our MT-VAE with the loss $\mathcal{L}_{AE}$. The ADAM optimizer (Kingma, 2014) with a constant learning rate of $1e-4$ is used. The batch size for training is set to be 4 and the number of total epochs is 100. The pretrained weights from image-based 2D VAEs (Rombach et al., 2022) used for image synthesis are employed for the initialization of the convolutional encoder and decoder, where the loaded weights of $E$, $D$ stay fixed during training. The parameters of the transformer $F$ and dense layers $\varphi$ are updated in this stage. No weight decaying schedule and data augmentation is used. For the designed temporal self-attention architecture, the number $M$ of stacked attention layers is set to be 8 and the hidden dimension is 128 in this study. The masking ratio $r$ is set as 0.6. In the second stage, the trained parameters in MT-VAE including $E$, $F$ and $\varphi$ are used to project data into latent space. In latent space, the denoising model is trained to predict less noisy samples by the 1:1 linear combination of $\mathcal{L}_{LDM}$ and $\mathcal{L}_{CA}$. The diffusion transformer (DiT) structure (Peebles & Xie, 2023) is used for constructing the denoising model with cascaded transformers. The ADAM optimizer with the same learning rate of $1e-4$ is used. The batch size is set to be 4 and the total epochs are 100. The parameters of the denoising model are the only learnable parameters in this stage. There are also no special weight decaying and augmentation schedules used. The number $N$ of cascaded DiTs is set to be 32 and the hidden size is 1152. These two stages are both performed on NVIDIA GeForce RTX 4090 with 24 GB $\times 4$. To inference with our PredLDM, the process is conducted and the timesteps $T$ for denoising is 1000.

## A.2 Additional Analysis on Critical Designs

For additional study of critical designs on KTH and SEVIR datasets, different settings of autoencoders including 3D VAEs, 2D VAEs + 1D convolutions and our MT-VAE are evaluated by pair-wise column correlation between synthetic data and real data. The value in heatmaps of correlation indicates the absolute divergence, where the more red area means the better correlation to real distributions. Meanwhile the performance calculated from predicted less noisy samples and ground truth during denoising processes with the trained denoising network with or without condition-aware reconstruction based constraints is plotted. The decoded predictions at different timesteps on these two datasets are also provided in this section.

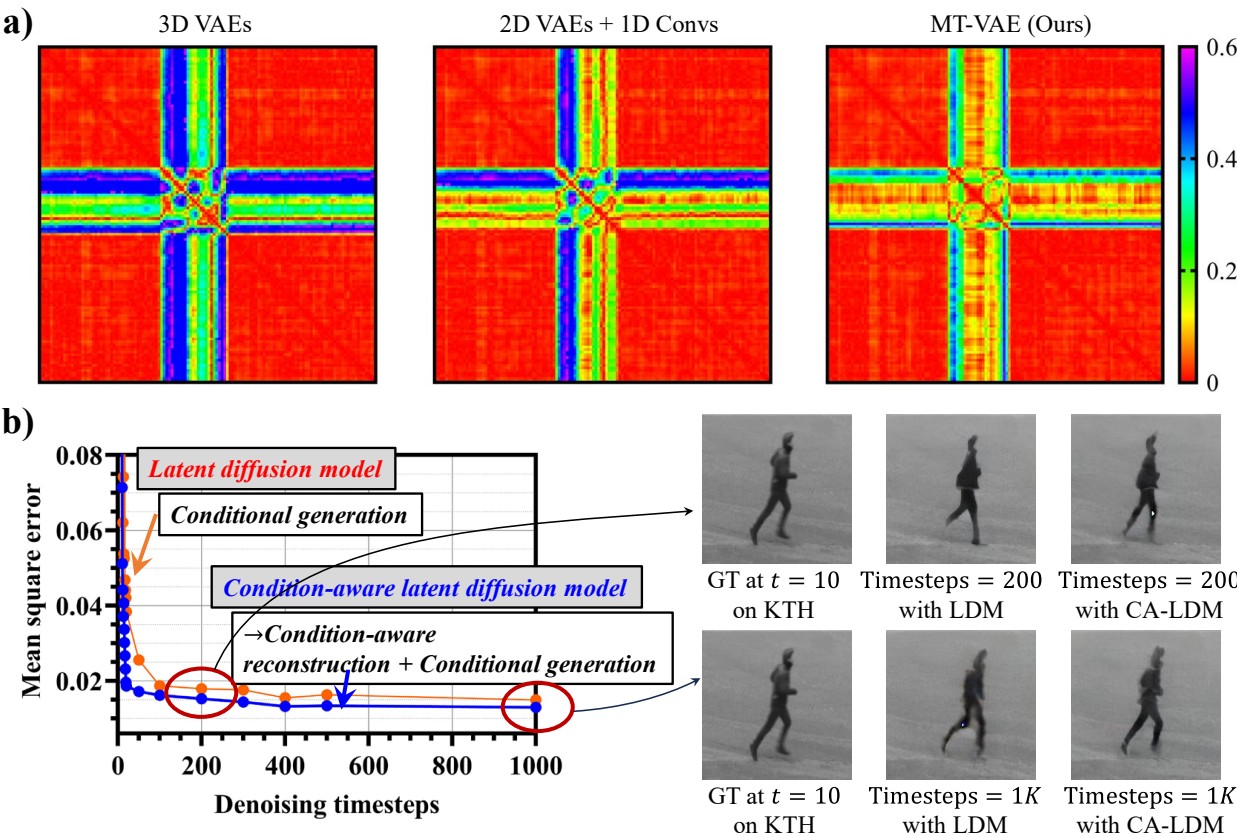

Figure 8: Additional ablation study on KTH. **a),** Heatmaps of correlation between synthetic data v.s. real data for MT-VAE. **b),** Influence of condition-aware latent diffusion. Performance in denoising processes is plotted on the left and the decoded predictions are on the right.

For analysis on KTH dataset, the results can be seen as in Figure 8. As the directions of human movement in this dataset are always vertical or horizontal and close to the center of images, the heatmaps of correlation reflect this characteristic. Results show that the synthetic distributions of MT-VAE are the closest to the real distribution. The setting of 2D VAEs + 1D convolutional temporal layers follows our setting and the setting of 3D VAEs behaves not competitive. This phenomenon shows that our MT-VAE is better at compressing spatiotemporal content into latent space with realistic distributions and our setting is effective. From the mean square error of sampled results during the denoising process, it is evident that the condition-aware reconstruction of CA-LDM is beneficial as the error of predictions from CA-LDM is much lower than the trained denoising model with only the original conditional generation related constraint. For the decoded visual results from timesteps at 200 and 1,000, the decoded visual quality from CA-LDM is better than the LDM setting and more timesteps are more conducive to producing detailed realistic visual appearance.

For analysis on SEVIR dataset, results are available as in Figure 9. The evolution of observations related to vertical liquid precipitation captured by the weather radar is distributed diversely in terms of physical geographic space, resulting the predicted dynamics difficult to be similar as ground truth, so heatmaps of correlation here are distributed in a disorderly manner. Results show that our setting of MT-VAE is still the best choice compared to other two settings, with highest relation to the distribution of real data. This again reveals that our MT-VAE is better at handling perceptual autoencoding. From the mean square error of sampled results during denoising processes, it can be seen that the false alarm rate of our predictions is much lower than predictions from the denoising model without condition-aware constraints. For the decoded visual results from different timesteps, the decoded predictions in this dataset reveal more accurate dynamics

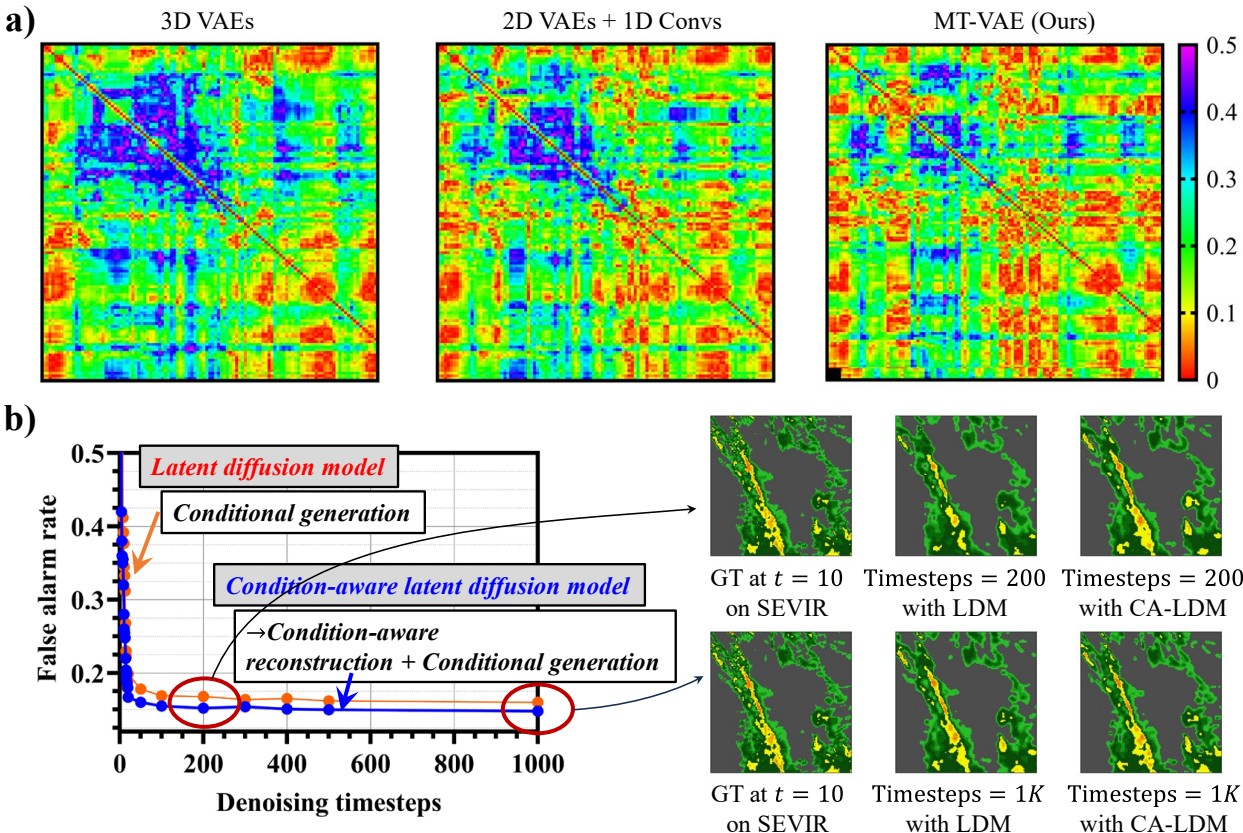

Figure 9: Additional ablation study on SEVIR. **a),** Heatmaps of correlation between synthetic data v.s. real data for MT-VAE. **b),** Influence of condition-aware latent diffusion. Performance in denoising processes is plotted on the left and the decoded predictions are on the right.

of heavy precipitation and realistic distributions of CA-LDM. Besides, realistic visual appearance with more details can be accessed by more denoising timesteps.

Table 4: Sensitivity analysis on the weighting scheme between conditional generation and condition-aware reconstruction on KittiCaltech. ↑ indicates the higher the better, whereas ↓ is the opposite. The best results are marked as bold.

| $\mathcal{L}_{LDM}$ | $\mathcal{L}_{CA}$ | SSIM ↑ | LPIPS ↓ | SSIM ↑ | LPIPS ↓ |
|---|---|---|---|---|---|
| 0.10 | 0.90 | 0.169 | 19.18 | 0.606 | 0.252 |
| 0.25 | 0.75 | 0.132 | 19.75 | 0.649 | 0.120 |
| 0.50 | 0.50 | **0.092** | **19.86** | **0.653** | **0.107** |
| 0.75 | 0.25 | 0.138 | 19.68 | 0.641 | 0.176 |
| 1.00 | 0.00 | 0.155 | 19.65 | 0.622 | 0.184 |

## A.3   More Results on Influence of Masking Ratios

To supplement the results in checking influence of masked modeling, we provide more results as in Figure 10. These results show the similar phenomenon as in experiments of the manuscript that the masked attention is necessary in our pipeline and the preferred setting is 0.6, although more settings like 0.65 are not tested, meaning more performance gain may lie in this ratio. Besides, for recommendations on tuning for new datasets, we suggest directly using a medium ratio as the first choice, evidenced by good performance in

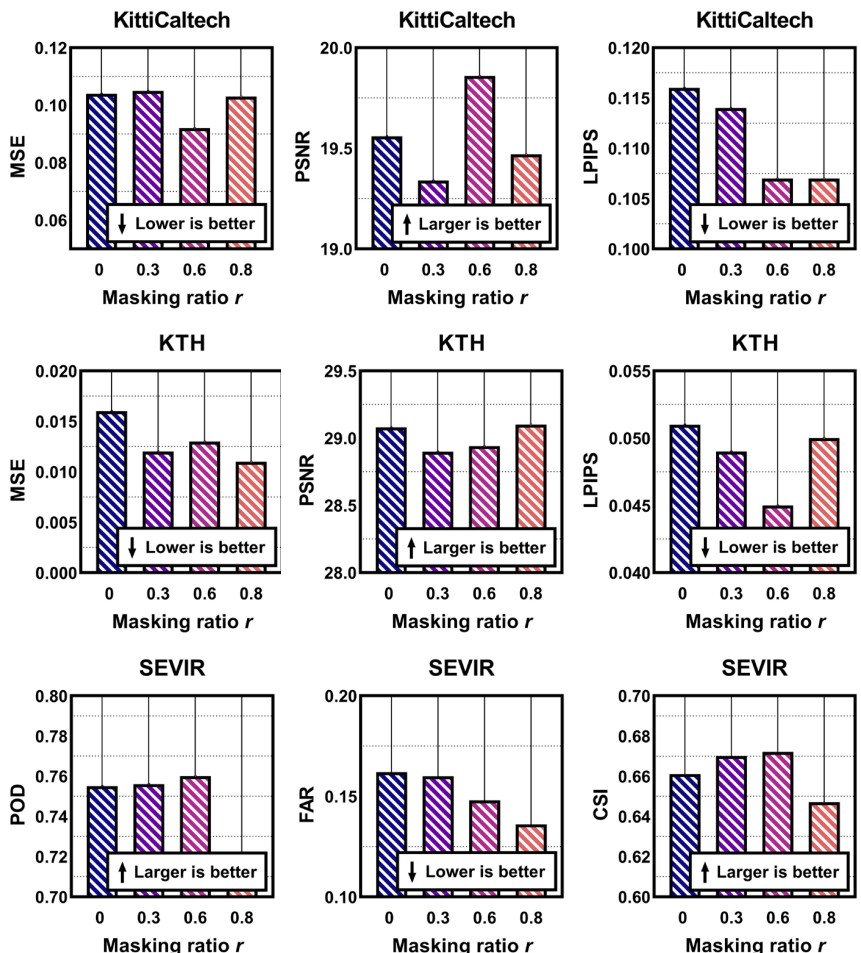

Figure 10: More results on influence of masking ratios. Remaining metrics on three datasets are presented.

Table 5: Computational cost on FLOPS, model parameters, inference time, training time and GPU memory.

|  | FLOPs (G) | Params (M) | Inference time (s/seq) | Training time (GPU hours) | GPU memory (GB) |
|---|---|---|---|---|---|
| ConvLSTM | 129.64 | 2.02 | 0.052 | 9.2 (1) / 2.3 (4) | 2.1 |
| PredRNN | 243.06 | 7.25 | 0.495 | 84.1 (1) / 21.0 (4) | 4.3 |
| MCVD | 5,434.28 | 367.60 | 1.620 | 646.1 (1) / 161.5 (4) | 8.9 |
| Ours | 4,566.57 | 421.26 | 1.331 | 530.4 (1) / 132.6 (4) | 14.8 |

our experiments on three different-occasion datasets. The alternative choice may be a larger one like 0.8, because for some simple occasions like KTH datasets a large ratio is still beneficial.

## A.4 Sensitivity Analysis on Weighting Scheme of Diffusion-based Loss

To investigate the sensitivity on the weighting scheme of diffusion-based loss functions, we conduct an additional experiment on weights of traditional diffusion-based conditional generation loss and condition-aware reconstruction-based loss. Results are reported in the Table 4. It needs to be mentioned that the setting for the conditional generation must not be 0, as this loss is critical to the task future prediction, so the minimum setting of this loss is 0.1. From experimental results on KittiCaltech, it appears that the setting of condition-aware loss is necessary. When the weight of $\mathcal{L}_{CA}$ becomes larger, scores on KittiClatech

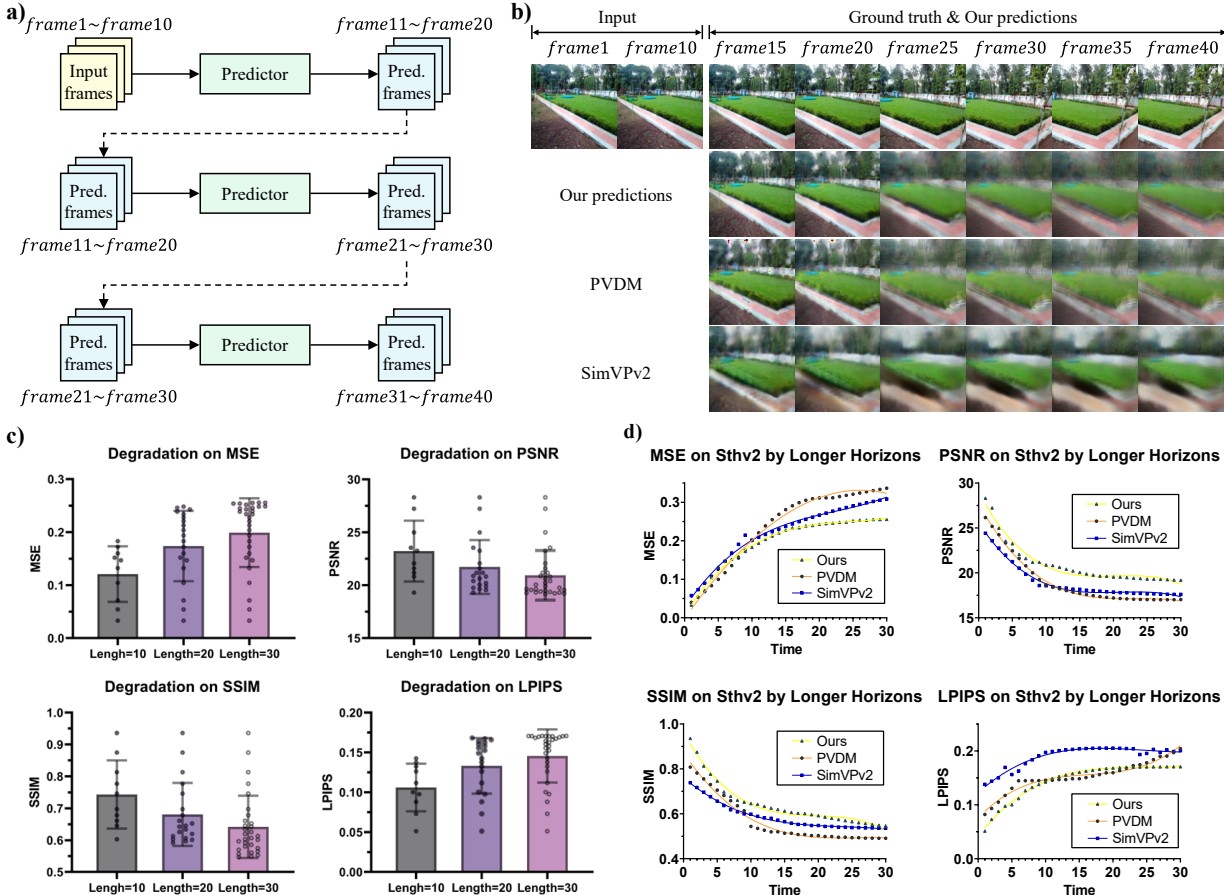

Figure 11: Additional analysis of our model at longer horizons and larger resolutions. Evaluation of our model on Sthv2 dataset is conducted as the inference manner in **a)**, iteratively predicting future frames. **b)**, Predictions of our model over longer time are visualized. Degration of preformance for our model is plotted as in **c)**, where MSE, PSNR, SSIM and LPIPS are evaluated. **d)**, Detailed comparison between competitive baselines and ours by longer horizons.

are more competitive. Besides, the setting of $\mathcal{L}_{LDM}$ should not be very small as this is the basic objective in spatiotemporal predictive tasks. It indicates that the linear combination of these loss functions is satisfied with the performance gain for common occasions.

## A.5 Analysis on Computational Cost

We report scores of ours and some baselines for better showing the cost as in below Table 5. Results show that the model scale and computational cost of classical predictive methods like ConvLSTM and PredRNN are attractive. But these two cost-related scores of ours are not very expensive compared to the diffusion baseline. Our inference speed is faster than MCVD, processing one sequence in 1.33 seconds. For training our architecture, firstly the MT-VAE is trained from scratch with around 39.3 GPU hours, around 9 hours by 4 Nvidia GeForce 4090 GPUs. For the diffusion process, i.e., the second training stage may cost 530.4 GPU hours, around 5.5 days with 4 Nvidia GeForce 4090 GPUs. When inferencing one sequence, 14.8 GB GPU memory is needed, it can be easily deployed on Nvidia GeForce 4090. Considering the compiled data here, we think the current setting is not quite real-time, where PredLDM produces 10 frames in 1.331 seconds, even with the compression ability from MT-VAE. If considering real-time application, acceleration skills are still needed.

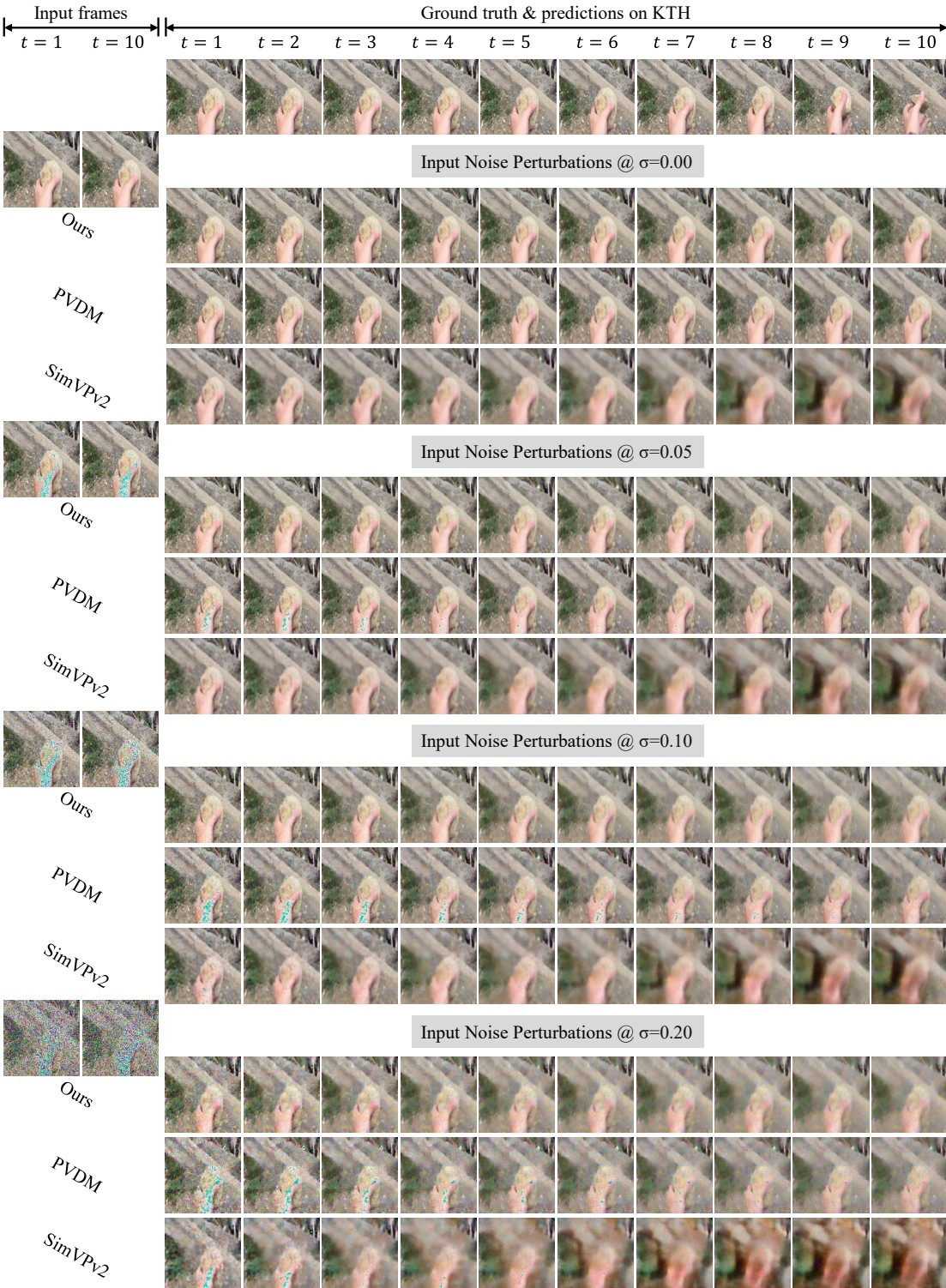

Figure 12: Robustness analysis by adding noise on input frames. Noise with different levels of standard deviation is added onto the input sequences. Predictive models including PVDM, SimVPv2 and ours are used to produce future sequences.

Table 6: MSE/MAE comparison between existing models and PredLDM on KittiCaltech, KTH as well as SEVIR datasets. ↑ indicates the higher the better, whereas ↓ is the opposite. The best results are marked as bold and the second best ones are marked as underline.

| Models | KittiCaltech | | KTH | | SEVIR | |
|---|---|---|---|---|---|---|
| | MSE ↓ | MAE ↓ | MSE ↓ | MAE ↓ | MSE ↓ | MAE ↓ |
| U-Net | 0.158 | 1.631 | 0.023 | 0.214 | 0.055 | 0.026 |
| PredRNN | 0.166 | 1.574 | 0.018 | 0.188 | 0.048 | 0.024 |
| PredRNN++ | 0.166 | 1.843 | 0.018 | 0.199 | 0.045 | 0.020 |
| ConvLSTM | 0.150 | 1.482 | 0.023 | 0.211 | 0.053 | 0.026 |
| MotionRNN | 0.151 | 1.605 | 0.015 | 0.168 | 0.042 | 0.023 |
| SimVP | 0.160 | 1.861 | 0.016 | **0.141** | 0.040 | **0.017** |
| SimVPv2 | 0.159 | 1.435 | 0.015 | 0.178 | 0.041 | 0.020 |
| MCVD | 0.125 | 1.277 | 0.027 | 0.277 | 0.068 | 0.036 |
| PVDM | 0.114 | **1.027** | 0.023 | 0.223 | 0.051 | 0.024 |
| PredLDM | **0.092** | 1.047 | **0.013** | 0.147 | **0.034** | 0.019 |

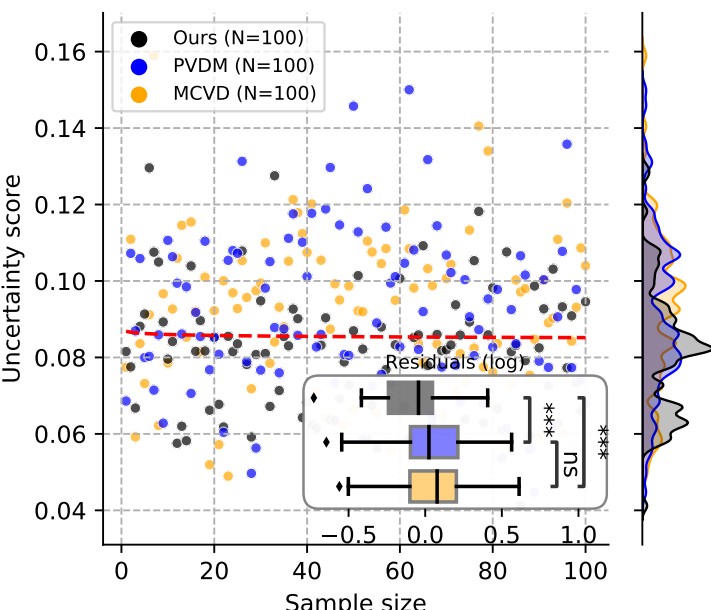

Figure 13: Uncertainty quality is quantified by standard deviation of ensemble predictions. 100 predictions are made by experimental models.

## A.6 Inference by Higher Resolutions and Longer Horizons

To evaluate by higher resolutions and longer horizons, we adopt an iterative inference strategy (Pathak et al., 2022; Bi et al., 2023). Given by 10 input frames, the model predicts the next 10 frames, which are then recursively fed back as input for subsequent predictions, as illustrated in Figure 11. Experimental results on the Sthv2 (Goyal et al., 2017) dataset where frames are processed at $256 \times 256$, revealing an acceptable decline in visual quality over extended time horizons, which is corroborated by a gradual degradation in quantitative metrics, including MSE, PSNR, SSIM, and LPIPS. From detailed temporal results compared by the competitive diffusion baseline PVDM and the deterministic baseline SimVPv2, our PredLDM shows better performance towards this experimental setting.

## A.7 More Discussion on MT-VAE and CA-LDM

Firstly, we discuss more on the masked attention in MT-VAE, related with masking methods compared to DiT (Ma et al., 2024), SVD (Blattmann et al., 2023a), Make-A-Video (Singer et al., 2022) and MCVD (Voleti et al., 2022). DiT (Ma et al., 2024) proposes two kinds of masking strategy noted by Only Video and Temporal Mask. For the first masking, they use simple masking from 0 or 1 on time tokens. For the second masking, they use frame-level one. Masking modeling is operated on the data space, on temporal tokens or the direct pixel of frames. SVD (Blattmann et al., 2023a) additionally exploits a binary mask where 1 indicates the presence of a conditioning frame and 0 refers to a mask embedding on the input of U-Net denoising network. We can see their masking is on the input of denoising network by concatenating a masking conditioning frame-size embedding. Make-A-Video (Singer et al., 2022) uses the image-based masking trained for frame interpolation, their masking is on data-level. MCVD (Voleti et al., 2022) masks frames in two ways including masking future/past frames, masking both past and future frames. It directly masks historical frames or future frames in data space. Different from these methods, our MT-VAE is operated on the attention matrix, especially on the scaled dot production matrix from query and key in transformers of VAE. Secondly, more discussion is conducted on CA-LDM. From many classical generation or prediction methods (Ma et al., 2024; Blattmann et al., 2023a; Singer et al., 2022; Voleti et al., 2022), they can all refer to the main architectures DDPM or DDIM. For ours, architectural novelty may be the CA-LDM by additional condition-aware reconstruction of historical observations. Another diffusion denoising network is also used to probabilistically modeling only on the condition, i.e., historical spatiotemporal sequences. Our diffusion part design consists both the commonly used conditional generation objection and importantly the probabilistic modeling of the condition itself. The reason we design this process in the denoising diffusion process is that different from existing T2V research, the condition of this task is raw historical spatiotemporal sequences while T2V methods use text as conditions, where text is high dimensional and our condition is more low-level and with plentiful structural details, more difficult to comprehend. So an additional diffusion denoising process is designed in our PredLDM.

## A.8 Robustness Analysis by Adding Noise

To substantiate the model's reliability real-world conditions where input sequences may be noisy, we perform the robustness analysis. Our evaluation introduces additive Gaussian noise with zero mean and controlled standard deviation at 0.00, 0.05, 0.10 and 0.20. To assess generalization under distribution shift, we conduct on Sthv2 by PVDM, SimVPv2 and our PredLDM. As illustrated in Figure 12, our model exhibits markedly superior output consistency compared to baselines across all noisy levels. These results provide superior empirical robustness for our model.

## A.9 Additional Results on MSE/MAE and Quantified Uncertainty

To enhance comparability, we provide more results on experimental analysis in MSE/MAE as Table 6. Furthermore, for our probabilistic model PredLDM, we provide a quantitative uncertainty analysis implemented by predictive standard deviation of ensemble predictions to rigorously evaluate its trustworthiness in critical applications on SEVIR in Figure 13. Results show the lower ensemble standard deviation primarily indicating higher predictability and lower intrinsic uncertainty for specific forecast cases.

## A.10 More Challenging Cases

Additional challenging cases are visualized with predictions by predictive models on KittiCaltech, KTH and SEVIR datasets. As in Figure 14 and Figure 15, the challenging cases on KittiCaltech are presented. The predictions show the accurate modeling capability in spatiotemporal variations on car driving scenes, with realistic visual appearance compared to other models where our predictions are of high fidelity and results of other models are in more blur over time. As in Figure 16 and Figure 17, the challenging cases on KTH dataset are available. In these challenging cases, we can witness that many models are in trouble of handling these cases, where small parts of image sequences appear significant variations of motion. Predictions from existing models are highly likely to generate blur, while the predictions of PredLDM are quite attractive

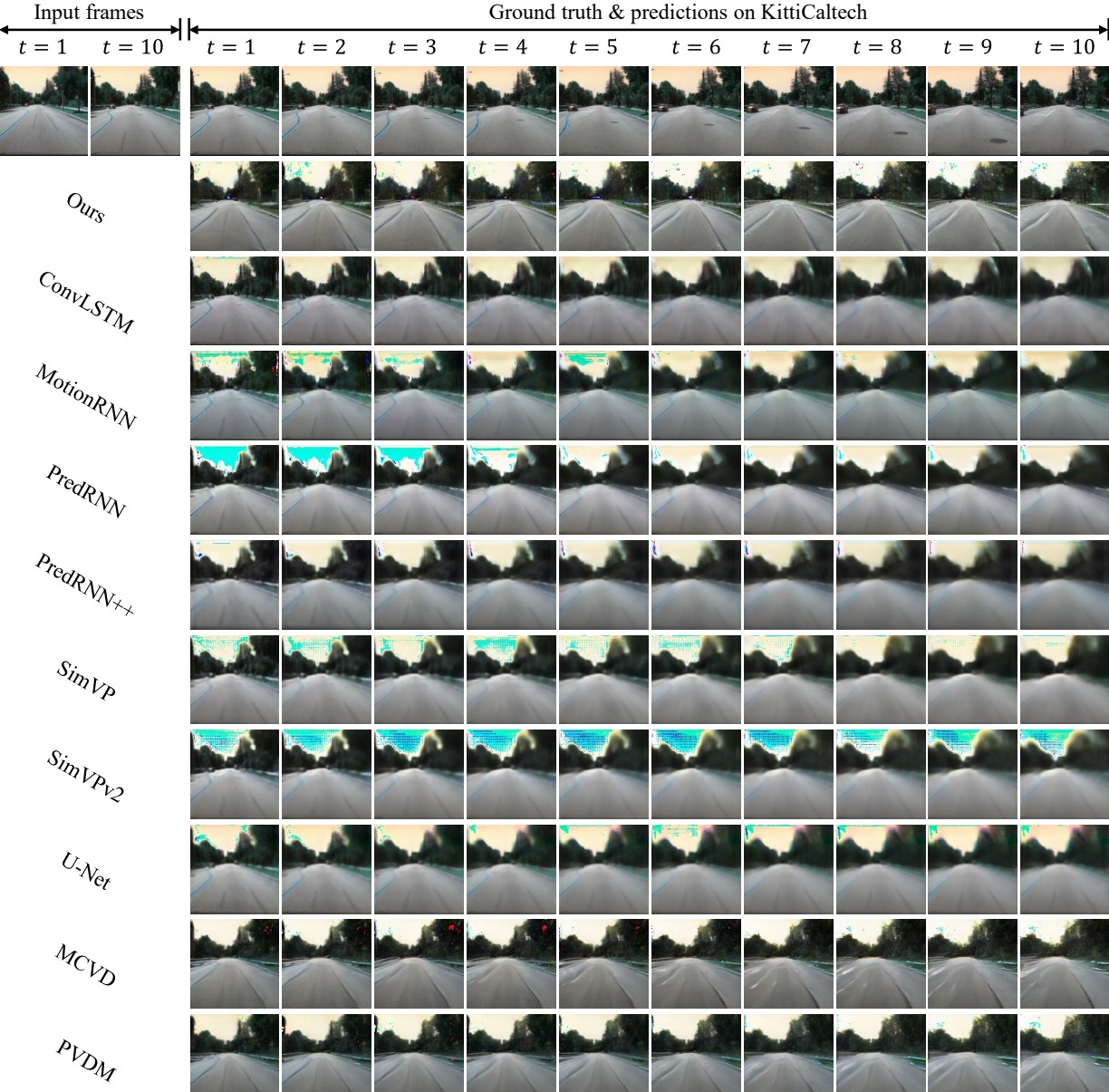

Figure 14: Challenging cases on KittiCaltech dataset. The first row refers to input spatiotemporal sequences and ground truth. The second row indicates sequences predicted by our PredLDM. Following rows are predictions from other models.

that they still appear realistic even to the last predicted frame. Besides, the better accuracy of predicted movement in these cases is evident for our model. As in Figure 18 and Figure 19, the challenging cases on SEVIR dataset are given. In weather pattern-related examples, existing predictive models are still likely to produce precipitation values with rough geophysical details, where the locations with high probabilities of rainfall are easy to omit, leading high risk for social failure preventing natural disasters. However, to results from PredLDM, predicted locations of heavy precipitation are more accurate and the overall distribution is more alike to the ground truth.

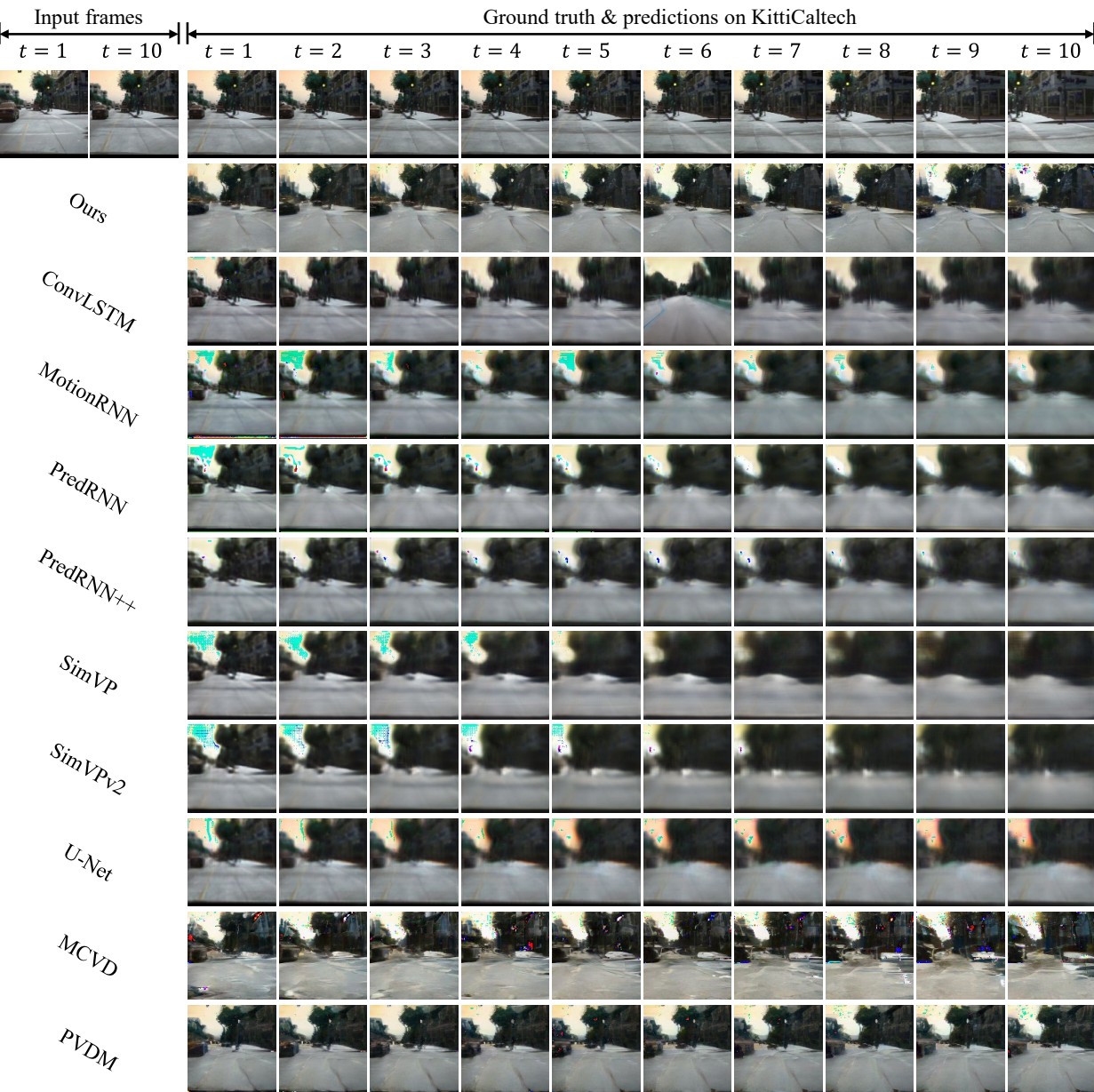

Figure 15: Challenging cases on KittiCaltech dataset. The first row refers to input spatiotemporal sequences and ground truth. The second row indicates sequences predicted by our PredLDM. Following rows are predictions from other models.

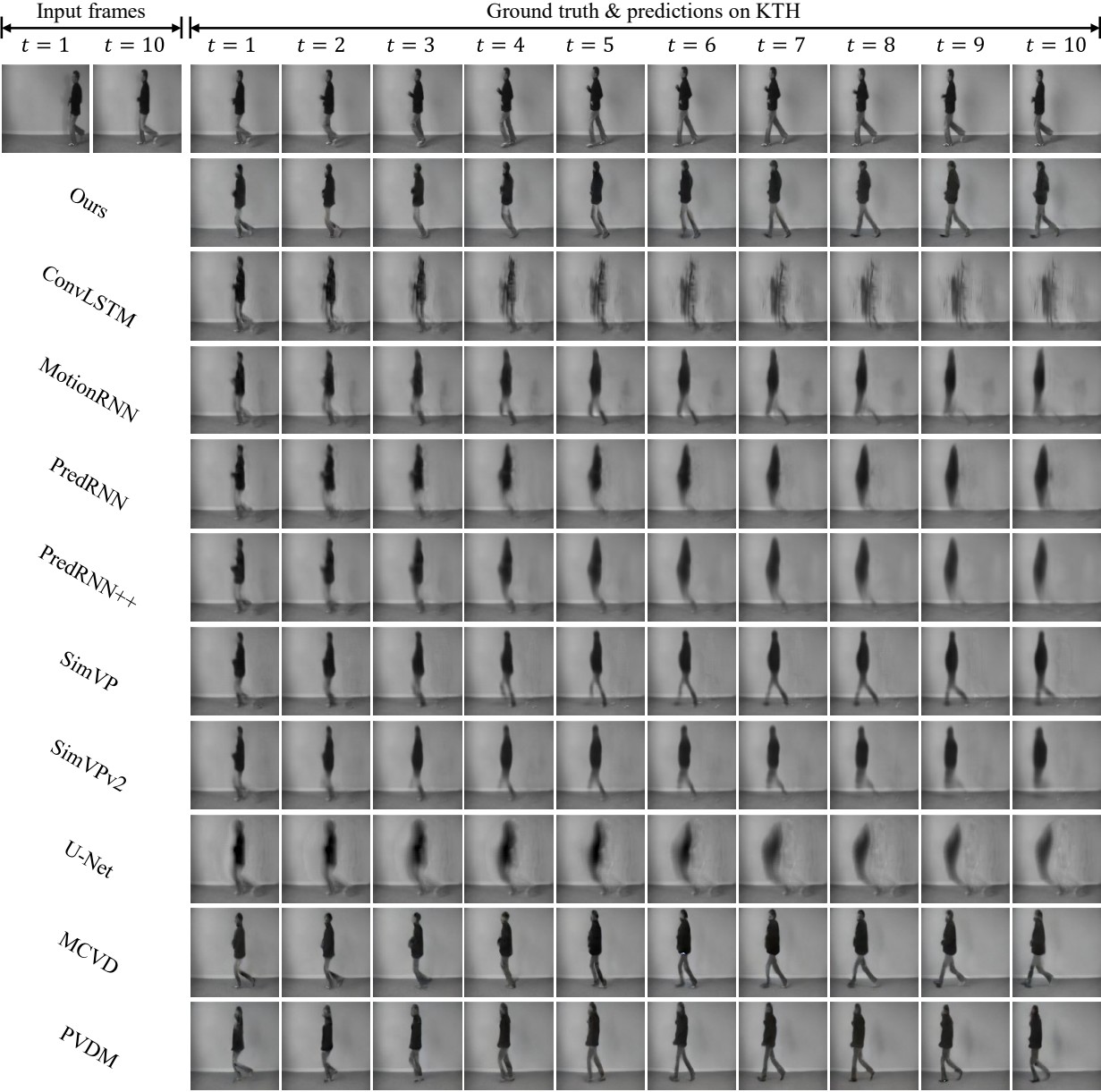

Figure 16: Challenging cases on KTH dataset. The first row refers to input spatiotemporal sequences and ground truth. The second row indicates sequences predicted by our PredLDM. Following rows are predictions from other models.

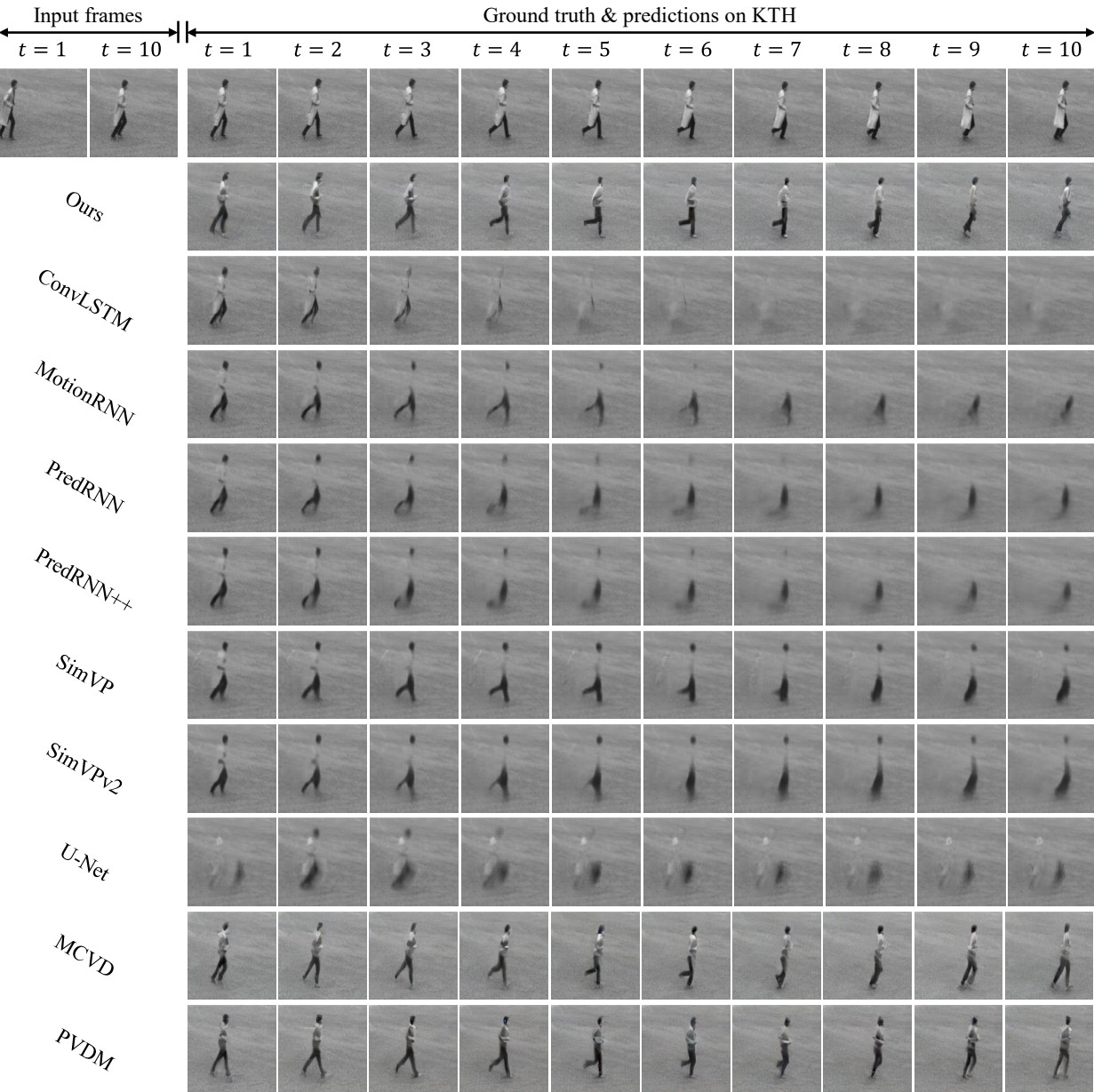

Figure 17: Challenging cases on KTH dataset. The first row refers to input spatiotemporal sequences and ground truth. The second row indicates sequences predicted by our PredLDM. Following rows are predictions from other models.

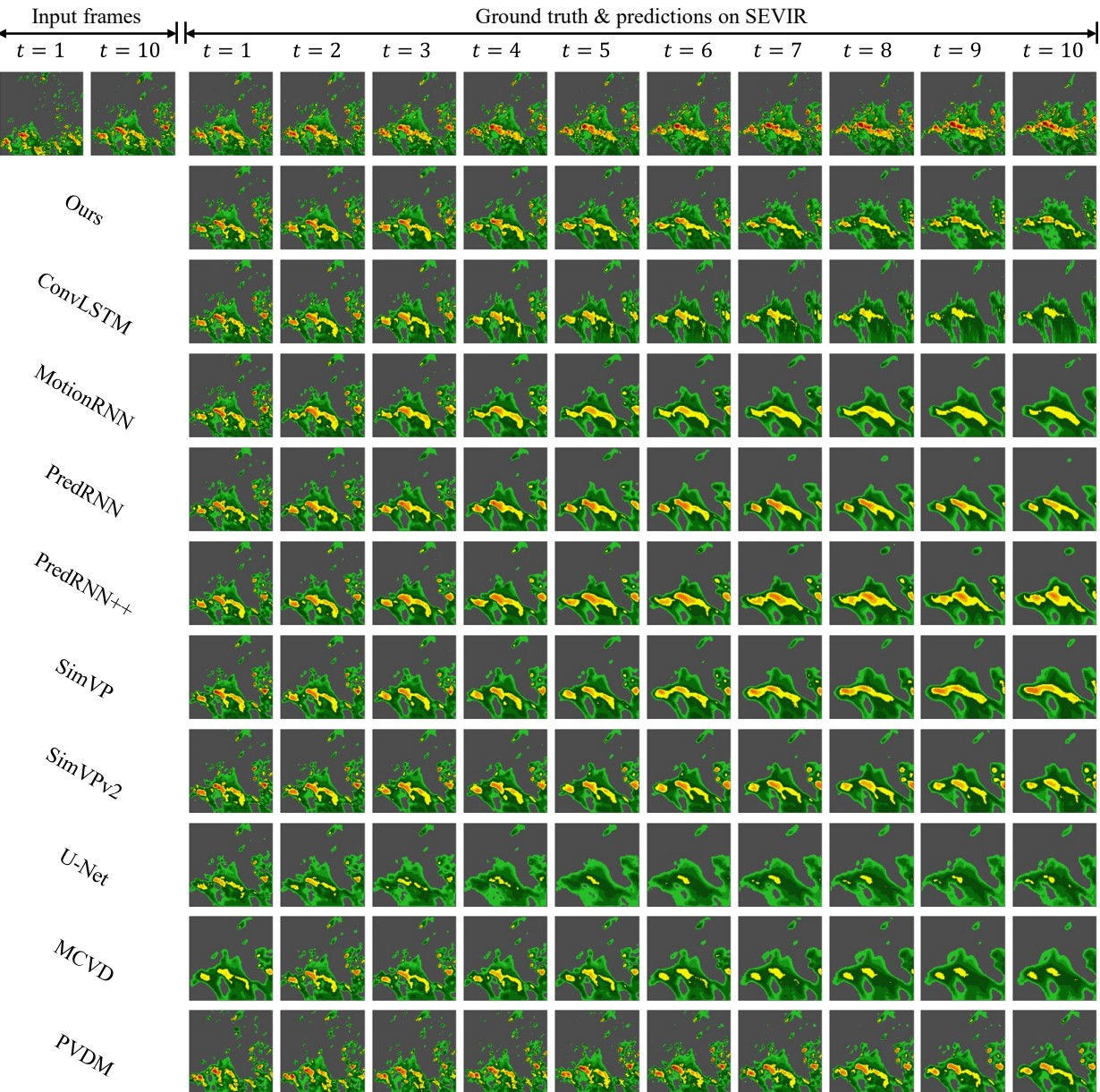

Figure 18: Challenging cases on SEVIR dataset. The first row refers to input spatiotemporal sequences and ground truth. The second row indicates sequences predicted by our PredLDM. Following rows are predictions from other models.

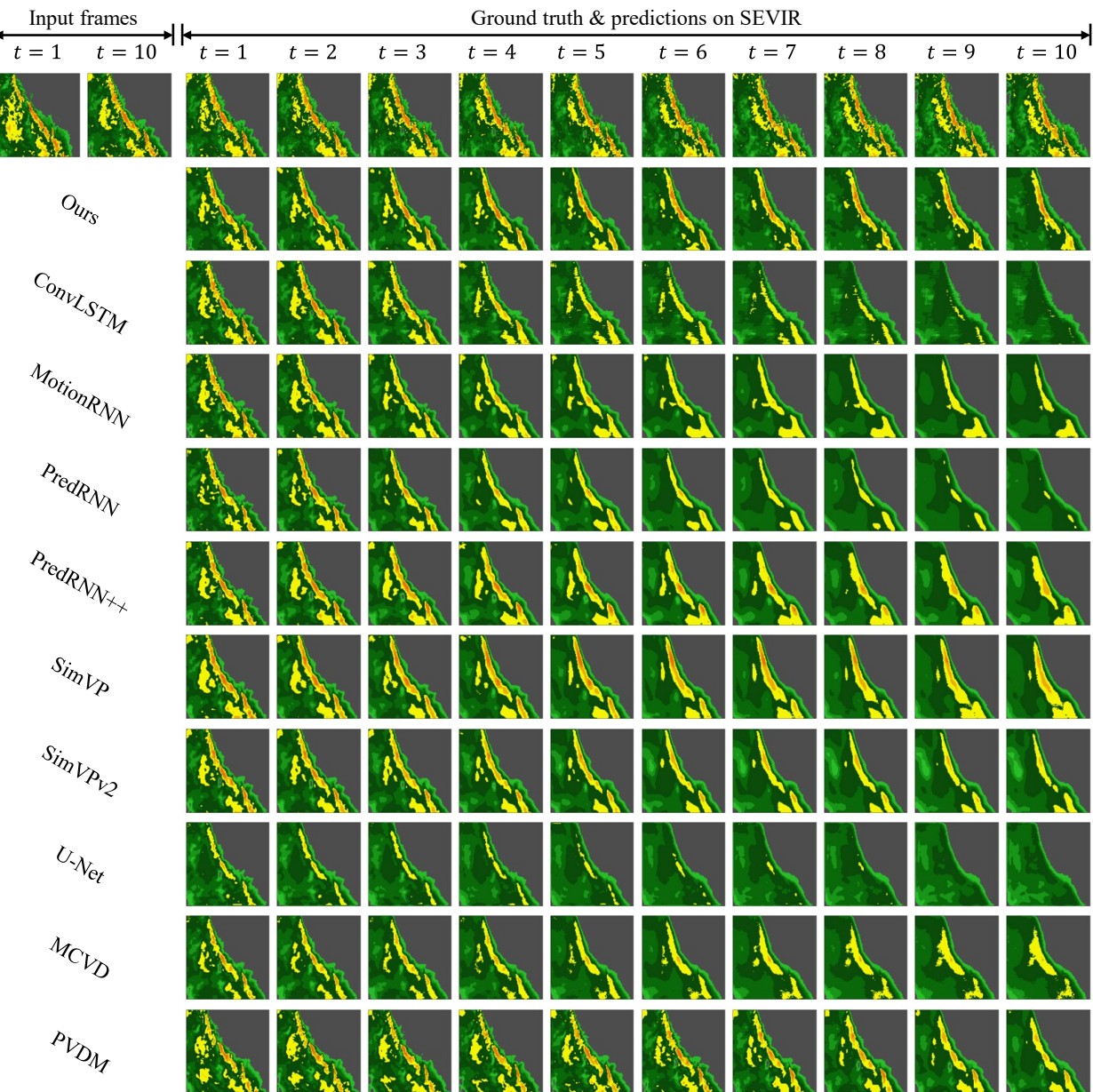

Figure 19: Challenging cases on SEVIR dataset. The first row refers to input spatiotemporal sequences and ground truth. The second row indicates sequences predicted by our PredLDM. Following rows are predictions from other models.

