# OpenReview forum: "PredLDM: Spatiotemporal Sequence Prediction with Latent Diffusion Models"
_TMLR — Accepted by TMLR_

### Review · Reviewer_aNY2 · 2025-10-30

**Summary Of Contributions:**

## (A) Brief Summary:

This submission introduces a spatiotemporal sequence prediction model PredLDM based on latent diffusion models (LDMs). It aims to address two key challenges: (1) capturing intricate global spatiotemporal coherence and (2) achieving a comprehensive understanding of historical image sequences, which serve as the condition.

This work proposes a two-stage framework to address this. The first stage uses a Masked-Attention Transformer-based VAE (MT-VAE) for perceptual compression. It employs a transformer-based masked self-attention to model global dependencies. The second stage introduces Condition-aware LDM (CA-LDM) to generate future latent sequences, conditioning on historical embeddings to bridge the gap between complex historical observation and future frame generation.

PredLDM is evaluated on KittiCaltech, KTH, and SEVIR datasets and show competitive results compared to baselines, especially in perceptual and distributional metrics. It contains quantitative benchmarks, thorough ablations, and qualitative analyses, suggesting improvements in visual fidelity and temporal consistency for challenging sequence prediction scenarios.

---
## (B) Strengths:

**(S1)** Clear problem formulation and motivation. Spatiotemporal sequence forecasting is important for real-world applications such as autonomous driving, robotics, and weather forecasting, each of which is included in this work’s experiments. It targets two key challenges in this field: global temporal coherence and history (condition) understanding, which are inadequately met by existing latent diffusion models like T2I and T2V.

**(S2)** Reasonable two-stage method design. To address the two identified issues, this paper directly maps its two main contributions (MT-VAE and CA-LDM) to them. It leverages both a transformer-based VAE for perceptual compression and a condition-aware latent diffusion model for spatiotemporal prediction. This is a reasonable combination that reflects existing research trends in high-fidelity generative modeling, as shown in Fig. 1.

**(S3)** Thorough quantitative and qualitative analysis. The experiments to support the claims are appreciated. It includes three diverse datasets (driving, human action, weather) with not only commonly-used frame/sequence-level metrics (PSNR, SSIM, LPIPS, POD, FAR, CSI) but also temporal dynamics evaluation (like those in Fig. 2 for time-evolving metrics) and distributional alignment (e.g., KVD/FVD in Fig 3). In addition, ablative studies in Tab 2–3 and Fig 5 show that effectiveness of individual components (MT-VAE, CA-LDM) in PredLDM.

**(S4)** Competitive results. PredLDM yields strong and consistent performance on KittiCaltech for driving, KTH for human activity, and SEVIR for weather. Notably. it achieves substantial gains in perceptual metrics like LPIPS (e.g., 30.5% on KittiCaltech) and relative error accumulation, as shown in Tab 1 and Fig 4. This provides strong quantitative evidence for the claim of generating realistic sequences.

**(S5) Visualization and error analysis.** Fig 4, 7, ablation heatmaps, and masking ratio studies in Fig 5, 6, 8, 9, 10 show careful error localization and condition-effect interpretation to support the claims of improved fidelity and robustness of PredLDM.

---
## (C) Weaknesses:

**(W1)** Ambiguous method illustration. It is vague on the precise method mechanics. For example, the "masked self-attention" in the MT-VAE is not clearly defined (e.g., is it input masking or an attention mask?) nor is its benefit for compression (rather than pre-training) explained. Similarly, the technical reason why the $\mathcal{L}_{CA}$ loss helps understanding is only described at a high level without precise justification.

**(W2)** Results discussion. For example, Tab 1 presents aggregate results, but there is little discussion of why some models like ConvLSTM outperform PredLDM in PSNR for certain datasets. Many similar points remain inadequately discussed, which I find regrettable given the extensive experiments. I recommend that the authors thoroughly discuss and derive insights based on existing experiment results, incorporating different methods’ designs. I believe this would be helpful for future research in spatiotemporal sequence prediction.

**(W3)** Some notation issues reduce readability. Some operators and function names are typeset in italics, which are commonly noted as upright (\text{} or \operatorname{}) to distinguish them from variables. Specifically, (i) “mAtt” in Eq. (5), (ii) “FeedForward” in Eq. (6), (iii) “softmax” in Eq. (9), (iv) “where” & “otherwise” in Eq. (10), (v) “Concat” in Eq. (11), and more. I recommend the authors thoroughly check the entire manuscript and revise all similar matters.

**(W4)** Incomplete literature review. There are several diffusion-based models specifically for spatiotemporal and time-series prediction, including Latent-Shift [1], PriSTI [2], Stochastic Diffusion [3], and TimeAutoDiff [4]. They are not discussed or compared against in the manuscript. While some of these are preprints on arXiv, they have accumulated a significant number of citations, indicating considerable influence in the field. Considering the method relevance, I recommend the authors incorporate related discussion in the revised manuscript.

**(W5)** Minor issues about ablation settings. MT-VAE is not compared against a non-masked transformer VAE. The CA loss is introduced with a 1:1 weighting to LDM loss without sensitivity analysis. I recommend the authors add related experiments to improve the robustness and academic rigor of this paper.

**(W6)** Figure quality and formatting. The presentation quality of figures is low. Many figures like Fig. 1, 3, 4, 6, 7  appear to be low-resolution bitmaps with noticeable zigzag artifacts, especially as the figures are mostly large in the manuscript. I strongly recommend the authors thoroughly check all the figures in both main text and appendix, replace them with high-quality vector graphics (by exporting SVGs directly to PDF) for better presentation. It would be with much lower file size than screenshot but with much better quality. Furthermore, for figure sub-labels (e.g., in Fig. 4), using bracketed labels like "a)," "b)," and "c)" is often clearer for identification than "a," "b," "c,". Disregard this suggestion if this is a recommended format from the TMLR official template.

**(W7)** Computational cost. This is a practical concern. The heavy two-stage model (Transformer VAE + 32-layer DiT) requires 1000 denoising steps for inference. There is no discussion of model parameters, training costs, or inference latency. IMHO, this is a very important aspect that is missing, especially in applications like autonomous driving where latency is crucial.

---

## Reference

[1] Latent-Shift: Latent Diffusion with Temporal Shift for Efficient Text-to-Video Generation, arXiv 2023.

- It introduces efficient temporal operations in latent diffusion for video/spatiotemporal generation, which is similar to this submission’s domain and design.

[2] PriSTI: A Conditional Diffusion Framework for Spatiotemporal Imputation, ICDE 2023.

- It proposes diffusion for spatiotemporal data imputation, underlying similar challenges in temporal modeling.

[3] Stochastic Diffusion: A Diffusion Probabilistic Model for Stochastic Time Series Forecasting, SIGKDD 2025.

- It is relevant as a model for uncertainty in spatiotemporal sequence prediction

[4] TimeAutoDiff: Combining Autoencoder and Diffusion Model for Time Series Tabular Data Synthesizing, arXiv 2024.

- It combines autoencoder and diffusion models for sequential data, which is relevant to this submission in architectures.

**Additional Comments:**

I hope my comments help fellow reviewers and editors understand the basis of my recommendation. I am open to follow-up discussions with the authors to help further strengthen this work and reach a consensus for the final decision.

**Audience:**

Yes

**Audience Explanation:**

This paper targets spatiotemporal sequence prediction, which is a fundamental task with broad real-world applications. The adaptation of diffusion models to this conditional generation problem is a very active and important research field. IMHO, the CA-LDM with $\mathcal{L}_{CA}$ loss in this work for improving conditionality is a design that researchers in generative modelling and video prediction would be interested in. The strong empirical results on perceptual metrics make it practical as a strong new baseline.

**Broader Impact Concerns:**

This work addresses generative video, but its application is constrained to short-term future prediction based on a historical context. It seems to pose no obvious ethical risks, and I have no broader impact concerns.

**Claims And Evidence:**

Yes

**Claims Explanation:**

The claims made in this submission is quite reserved. It stated that PredLDM produces more "realistic predictions" than existing model. This is well-supported. First, it provides convincing evidence through competitive empirical results on the LPIPS, FVD, and KVD metrics, which are more suitable for measuring realism than traditional MSE/PSNR. Second, the qualitative results in Fig. 4, 7, and 11-16 further substantiate this, which shows a clear visual improvement in sharpness and detail retention over time.

However, the claim that MT-VAE successfully compresses "intricate global coherent spatiotemporal content" is supported weakly. Tab. 2 ablation shows it outperforms 3D VAEs and 2D VAEs + 1D Convs, but there is no ablation against a standard, non-masked transformer VAE. Thus, it is unclear if the "masked" part of the contribution is meaningful.

**Requested Changes:**

Most of my concerns and corresponding requested changes and suggestions have been stated in the Weaknesses section. Overall, I have a positive view on this submission. I encourage the authors to focus their efforts on addressing those points, especially **(W1)** **(W3)** **(W5)** **(W6)**, as they are critical for securing recommendation for acceptance.

The following are more specific and open questions to help the authors think more deeply about certain design choices and experiment setups, which I hope could be helpful for this and future work:

**(Q1)** What is the impact of OOD (out-of-distribution) or distribution-shift scenarios (e.g., unusual traffic/weather patterns) on PredLDM’s predictive quality and calibration? Would the model’s performance degrade smoothly, or does it show catastrophic failures? Some discussion or targeted experiments would be valuable.

**(Q2)** Could the authors clarify design choices regarding masking ratio in Fig 6, 10, and provide practical heuristics for tuning in new domains/datasets? It would also be helpful for the community to use PredLDM in other scenarios.

**(Q3)** How does the length of the input and output sequences affect the model's performance? Can PredLDM easily adjust to different forecasting horizons? Are there any modifications necessary for very long or short sequence prediction tasks?

---

> ### Author Response · Authors · 2025-11-27
> **Respond to Review of Paper5927 by Reviewer aNY2**
>
> **(W1)** Improving method illustration on masked self-attention, precise justification for compression and condition-aware reconstruction.
>
> **Response**: Thank you for your valuable instruction better clarifying our work. We have made modifications in the modified pdf submission following below detailed response.
>
> - We are pleased at clarifying that masked self-attention is applied on the attention matrix. Specifically, for clarifying the exact masked attention mechanism in our work, we start telling the process of calculating the scaled dot production of self-attention, the masking is operated on the attention matrix via equation (9) by $M$, where we can see this masking modulating the scaled dot production between Query and Key. The definition of this masking matrix $M$ is in equation (10), where $md(j)$ is the random binarized output with the masking ratio of the same size as time length and $j$ indicates any time location in $M$. So, $M_{j}$ refers to zero or negative infinite values-based vectors as in equation (10), and $M_{j}$ constructs $M$, $M_{j} \in M$. Here is the exact process masked self-attention.
>
> - For compression, varying spatiotemporal patterns can be accessed by our MT-VAE, as the masking operation creates more attention matrix-based samples along time dimensions. The attention matrix in our work is accessed from dense convolutions from previous latent input samples. By creating more intermediate attention feature samples, diverse spatiotemporal patterns can be observed during compression. We call this perceptual compression as more diverse data can be compressed by one observation, so many complex fractions of the same piece of time sequences can be linked tightly. This is the purpose of designing MT-VAE.
>
> - For better clarifying $L_{CA}$, we start from the angle of maximization mutual information between condition variables and outcomes. Different from classical DDPM or DDIM architectures, ours CA-LDM owns additional condition-aware reconstruction of historical observations. So our diffusion part design consists of both the commonly used conditional generation objection and importantly the probabilistic modeling of the condition itself. The reason we design this process in the denoising diffusion process is that different from existing T2V research, the condition of this task is raw historical spatiotemporal sequences while T2V methods use text as conditions, where text is high dimensional and our condition is more low-level and with plentiful structural details, more difficult to comprehend. As a result, in order to maximize the mutual information beyond joint distribution $p\left( z|c \right)$, we construct the condition-aware denoising diffusion loss by adding $p(c)$ during denoising steps.
>
> **(W2)** Supplementing discussion on experimental results.
>
> **Respond**: Following your valuable and encouraging suggestions, we add a subsection '**Discussion and Limitation**' for better discussing our experimental results. Several key points are inspired and have been reported, especially the suggested phenomenon.
>
> - Distortion-perception trade-off. We have observed a phenomenon from our experimental results especially in Table 1 that some classical predictive models such as ConvLSTM and MotionRNN behave excellent performance in structured-based metrics PSNR and SSIM, while not good at perceptual metric LPIPS. This states the trade-off in this context. Distortion (measured by PSNR and SSIM) and perceptual quality (LPIPS) are at odds with each other. Through our experiments, it can be found out that classical deterministic models excel in PSNR/SSIM because they are optimized for pixel-level fidelity, while probabilistic ones excel in perceptual metrics because they are optimized for statistical realism and diversity of outputs. This reveals the selection preference for specific downstream usage. For example, in some accuracy required occasions for example scientific measurement, the easy-implement and accurate classical models are preferred, while for some realistic necessary usages, probabilistic modeling may be an excellent choice.
>
> - Time-evolving performance. For autoregressive models like ConvLSTM, PredRNN and MotionRNN, a process generating frames or tokens one at a time. For sequence2sequence models like U-Net and SimVP, the whole spatiotemporal sequence is predicted by deterministic optimization. For diffusion-based models, the denoising process runs for a predefined number of steps to predict results from noise. From results in Figure 2, Figure 3 and Figure 4, it can be observed that the diffusion-based modeling is beneficial to reduce time iterative error. Besides, the proposed PredLDM can serve as a competitive baseline in this field.

---

> ### Author Response · Authors · 2025-11-27
> **Respond to Review of Paper5927 by Reviewer aNY2 - Continued Part 2**
>
> - Masked operations in attention matrix are beneficial to temporal modeling. DiT \[1\] proposes two kinds of masking strategies, including simple masking from one-hot notes on time tokens and frame-level masking, which is operated on the direct pixel of frames. SVD \[2\] additionally exploits a binary mask where 1 indicates the presence of a conditioning frame and 0 that of a mask embedding on the input of U-Net denoising network. We can see their masking is on the input of denoising network by concatenating a masking conditioning frame-size embedding. Make-A-Video \[3\] uses the image-based masking trained for frame interpolation, still limited on data-level. MCVD \[4\] masks frames in two ways on both past and future frames. Different from these methods, our MT-VAE is operated on the attention matrix, especially on the scaled dot production matrix from query and key in transformers of VAE. From results in Figure 6 and Figure 10, it can be observed that our masked modeling is also beneficial against the version without masking (when the masking ratio is 0). For future research, more deep research on attention matrix-based masked modeling may be of much value.
>
> 1: Xin Ma, et al. Latte: Latent diffusion transformer for video
> generation. arXiv preprint arXiv:2401.03048 (2024).
>
> 2: Blattmann et al. Stable video diffusion: Scaling latent video
> diffusion models to large datasets. arXiv preprint arXiv:2311.15127
> (2023).
>
> 3: Singer et al. Make-a-video: Text-to-video generation without
> text-video data. arXiv preprint arXiv:2209.14792 (2022).
>
> 4: Voleti et al. Mcvd-masked conditional video diffusion for prediction,
> generation, and interpolation. Advances in neural information processing
> systems 35 (2022), 23371--23385.
>
> **(W3)** Solving notation issues by distinguishing between variables and operators.
>
> **Respond**: Following your professional instructions, we have modified and checked corresponding notion issues for enhancing readability. \\operatorname{} is used to make operators as upright in Equation 5-11, as well as corresponding descriptions.
>
> **(W4)** Improving literature discussion by incorporating recommended citations.
>
> **Respond**: Following your instructions, suggested citations are added for better discussion on diffusion-related work. They are cited as below in Related Works.
>
> - ... autoencoders to learn an expressive prior over discretized latent space. This autoencoder-diffusion paradigm has also been effectively adapted for sequential data synthesis, as demonstrated in TimeAutoDiff (Suh et al., 2024). In the second stage, a denoising network commonly implemented by U-Net (Ho et al., 2020; Ronneberger et al., 2015; Dhariwal & Nichol, 2021) is trained to predict the less noisy video samples progressively, reversing the diffusion process where Gaussian noise is iteratively added onto the raw data with predefined timesteps. Concurrently, efficient temporal modeling techniques like those in Latent-Shift (An et al., 2023) have advanced video generation by introducing specialized operations in the latent space. Beyond generation tasks, PriSTI (Liu et al., 2023b) addresses missing data challenges through conditional diffusion frameworks. Similarly, Stochastic Diffusion (Liu et al., 2024a) extends this capability to uncertainty-aware time series forecasting, highlighting the versatility of diffusion models in handling complex temporal dependencies. Different form T2I and T2V models, the condition of our PredLDM is historical observations. This difference makes the condition is not as easy as text prompts used to be, as the spatiotemporal content is low dimensional and highly complex.
>
> **(W5)** Solving issues about ablation settings on masking and weighting balance.
>
> **Respond**: Thank you for providing professional suggestions supplementing our work on non-masked comparison and weighting analysis. Following your instructions, we have made corresponding modifications and more effort is added into the revised version.
>
> - To enhance the comparison against the non-masked transformer-based setting in VAE, we add more discussion in the subsection '**Influence on Masking Ratios**' as: For investigating the influence of the masking ratio r on forecasting performance on three datasets, different settings of masking ratios are experimented as in Figure 6. More results are provided as in Supplementary Section A.3. Firstly, when the masking ratio is 0, the setting refers to the masking mechanism does not work in MT-VAE and instead the non-masked transformer-based VAE compresses spatiotemporal data. Comparing against the non-masked settings, it can be witnessed that the masked attention in MT-VAE is effective in most aspects on three datasets. Meanwhile, these results also indicate that the setting of masking ratio is preferred as 0.6 in our experiments.

---

> ### Author Response · Authors · 2025-11-27
> **Respond to Review of Paper5927 by Reviewer aNY2 - Continued Part 3**
>
> - To investigate the suggested sensitivity from 1:1 weighting to LDM
>   loss, we conduct an additional experiment on the weighting scheme of
>   weights on traditional diffusion loss and condition-aware
>   reconstruction-based loss. Added experimental results and discussion
>   are placed in the revised manuscript. In details, results are reported
>   in the table below. The setting for the conditional generation must
>   not be 0, as this loss is critical to the task future prediction, so
>   the minimum setting of this loss is 0.1. From experimental results on
>   KittiCaltech, it appears that the setting of condition-aware loss is
>   necessary. When the weight of $L_{CA}$ becomes larger, scores on
>   KittiClatech are more competitive. Besides, the setting of $L_{LDM}$
>   should not be very small as this is the basic objective in
>   spatiotemporal predictive tasks. It indicates that the linear
>   combination of these loss functions is satisfied with the performance
>   gain for common occasions.
>
> - A subsection named '**A.5 Sensitivity Analysis on Weighting Scheme of
>   Diffusion-based Loss**' is added in the supplementary part of the
>   modified version.
>
> Table. Sensitivity analysis on the weighting scheme between conditional
> generation loss and condition-aware reconstruction loss on KittiCaltech.
>
> | $L_{LDM}$  | $L_{CA}$  | MSE   |       PSNR    |     SSIM       |  LPIPS|
> |:-------:|:----------:|:-------:|:-------:|:-------:|:-------:|
> |  0.1   |        0.9    |      0.169    |    19.18    |    0.606   |     0.252|
>  | 0.25    |      0.75     |    0.132   |     19.75   |     0.649     |   0.120|
>   |0.5     |      0.5     |     0.092    |    19.86    |    0.653    |    0.107|
>   |0.75    |      0.25    |     0.138    |    19.68    |    0.641     |   0.176|
>   |1       |      0     |       0.155  |      19.65     |   0.622     |   0.184|
>
> **(W6)** Improving figure quality and formatting.
>
> **Respond**: Thank you for your professional suggestions on improving
> the quality of display in our work. Following your instructions, firstly
> \"a),\" \"b),\" and \"c)\" are used to display \"a,\" \"b,\" \"c,\" in
> all figures. Secondly, all figures are transformed as pdf for improving
> the visual quality of our displayed figures.
>
> **(W7)** Adding analysis on computational cost.
>
> **Respond**: Thank you for your professional suggestion. Following your
> instruction, we have compiled the cost-related scores like you suggested
> model parameters, training costs or inference latency.
>
> - We report scores of ours and some baselines for better showing the
>   cost as in below Table. The results show that the model scale and
>   computational cost of classical predictive methods like ConvLSTM and
>   PredRNN are attractive. But these two cost-related scores of ours are
>   not very expensive compared to the diffusion baseline. Our inference
>   speed is faster than MCVD, processing one sequence in 1.33 seconds.
>   For training our architecture, firstly the MT-VAE is trained from
>   scratch with around 39.3 GPU hours, around 9 hours by 4 Nvidia GeForce
>   4090 GPUs. For the diffusion process, i.e., the second training stage
>   may cost 530.4 GPU hours, around 5.5 days with 4 Nvidia GeForce 4090
>   GPUs. When inferencing one sequence, 14.8 GB GPU memory is needed, it
>   can be easily deployed on Nvidia GeForce 4090. Considering the
>   compiled data here, we think the current setting is not quite
>   real-time, where PredLDM produces 10 frames in 1.331 seconds, even
>   with the compression ability from MT-VAE. If considering real-time
>   application, acceleration skills are still needed.
>
> - These results and discussion are added into our modified version, as a
>   subsection in '**A.6 Analysis on Computational Cost**'.
>
> - Added content in subsection subsection '**4.9 Discussion and
>   Limitation**', as 'Real-time performance. While our model demonstrates
>   competitive performance, it is important to acknowledge its
>   computational limitations. Although the model scale and computational
>   cost are not excessive compared to the diffusion baseline (MCVD) and
>   our inference speed of 1.33 seconds per sequence is faster, the
>   current setting is not ideal yet for real-world fast applications
>   satisfying real-time usage such as autonomous driving. Specifically,
>   generating 10 frames takes 1.331 seconds, even with the compression
>   provided by the MT-VAE. Therefore, for practical real-time deployment,
>   further acceleration techniques would be necessary. For offline usages
>   like urban supervision, analyzing or weather forecasting, it may
>   remain capable of serving.'

---

> > ### Author Response · Authors · 2025-11-27
> > **Respond to Review of Paper5927 by Reviewer aNY2 - Continued Part 4**
> >
> > **Table. Computational cost on FLOPS, model parameters, inference time,
> > training time and GPU memory.**
> >
> >  |      |        FLOPs (G)  |   Params (M)  |  Inference time (s/seq)  |  Training time (GPU hours)  |   GPU memory (GB)|
> >   |:-------:|:----------:|:-------:|:-------:|:-------:|:-------:|
> >    |ConvLSTM   | 129.64   |  2.02   |    0.052   |   9.2 (1) / 2.3 (4)  |    2.1|
> >   | PredRNN     |243.06  |   7.25     |  0.495    |   84.1 (1) / 21.0 (4) |     4.3|
> >    |  MCVD    | 5,434.28 |  367.60    |  1.620  |    646.1 (1) / 161.5 (4) |    8.9|
> >    |  Ours   |  4,566.57  | 421.26   |   1.331   |   530.4 (1) / 132.6 (4) |   14.8|
> >
> > **(Q1)** Discussing on the impact of OOD and conducting experiments on
> > the added HKO-7 dataset.
> >
> > **Respond**: Thank you for providing this valuable question. We have
> > made some attempts which we adapted our trained model on SEVIR into
> > HKO-7 dataset which is a weather nowcasting dataset with different
> > distributions. Although we haven't enough time to conduct extensive
> > experiments on the OOD issue, we would like to share the phenomenon that
> > this adaptation without any finetuning would cause at least 10% or more
> > performance degradation, compared to ours trained directly on this OOD
> > dataset. It may arise from the limited scale of training data, little
> > adaptation from other knowledge or being too dependent on the data
> > driven mechanism. We think this OOD issue is the excellent research
> > direction for our work, it is an inspiring question and we would
> > consider this as an important angle in our future work.
> >
> > **(Q2)** Clarifying design choices regarding masking ratio and how to
> > tune for new dataset.
> >
> > **Respond**: Thank you for your helpful issue and we are pleased to
> > share some practical heuristics for masking ratios and tuning in new
> > domains. Firstly, we'd like to share about the masking ratio. The
> > masking ratio-based setting design between 0, 0.3, 0.6 and 0.8 are
> > expected to observe the performance behaviors when the masking not
> > operated goes forwards more masking proportions. 0 refers to the
> > non-masked transformer-based VAE and 0.8 means most of data is masked,
> > they indicate the two opposites. From results in Fig. 6 and Fig. 10, it
> > shows that an appropriate ratio affects the performance importantly. For
> > our experiments, a medium ratio like 0.6 would most benefit our model,
> > although more settings like 0.65 are not tested, meaning more
> > performance gain may lie in this ratio. Secondly, for recommendations on
> > tuning for new datasets, we suggest directly using a medium ratio as the
> > first choice, evidenced by good performance in our experiments on three
> > different-occasion datasets. Alternative choices may be a larger one
> > like 0.8, because for some simple occasions like KTH datasets a large
> > ratio is still beneficial. These suggestions are reflected in the
> > revised version of our manuscript.
> >
> > **(Q3)** Studying influence by the length of input and output.
> >
> > **Respond**: A subsection named by '**A.7 Inference by Higher
> > Resolutions and Longer Horizons'** is added in the modified version. To
> > evaluate by higher resolutions and longer horizons, we adopt an
> > iterative inference strategy. Given by 10 input frames, the model
> > predicts the next 10 frames, which are then recursively fed back as
> > input for subsequent predictions, as illustrated in Figure 11.
> > Experimental results on the Sthv2 dataset (which is an added dataset),
> > revealing an acceptable decline in visual quality over extended time
> > horizons, which is corroborated by a gradual degradation in quantitative
> > metrics, including MSE, PSNR, SSIM, and LPIPS. For predicting the
> > variable length of output sequences, all shorter predictions can
> > directly be accessed from results in longer sequences, while longer
> > horizons are available by this iterative inference strategy.
> >
> > Overall, we hope our effort may help. Best wishes.

---

> > > ### Comment · Reviewer_aNY2 · 2025-12-16
> > > **Official Response by Reviewer aNY2**
> > >
> > > I want to thank the authors for their precise and thorough rebuttal. I appreciate the significant effort put into conducting additional experiments, revising the text, and updating the visualizations.
> > >
> > > In brief, the authors have addressed my main concerns on below points:
> > >
> > > * Method Clarity (W1 & W3): The clarification regarding masked self-attention mechanism (specifically Eq. 9 and 10) and the precise definition of the masking matrix has resolved the ambiguity in the first submission. The notation fixes (using \operatorname) have improved readability.
> > >
> > > * Baselines and Ablations (W5): The inclusion of non-masked transformer VAE baseline and sensitivity analysis for the loss weighting scheme provides necessary empirical rigor that was previously missing. This strengthens the justification for the model architecture choices.
> > >
> > > * Computational Analysis (W7): Section A.6 and the accompanying table showing FLOPs, parameters, and inference latency are crucial additions. While the method is computationally heavier than simple baselines, the transparency regarding these costs allows for a fair evaluation of the trade-off between fidelity and efficiency.
> > >
> > > * Discussion and Literature (W2 & W4): The expanded literature review (incorporating diffusion forecasting works) and the added discussion on the distortion-perception trade-off adequately contextualize the results, explaining why deterministic metrics (PSNR) might lag while perceptual quality excels.
> > >
> > > * Robustness and Scalability (Q1 & Q3): I acknowledge the preliminary results on the HKO-7 dataset and the iterative inference experiments on Sthv2. While fully solving OOD robustness is outside the immediate scope of this paper, the authors' acknowledgment and the current results are sufficient for publication.
> > >
> > > Overall, the rebuttal has improved the technical depth and presentation quality of the manuscript. It presents a technically sound approach with convincing evidence supporting its claims on spatiotemporal consistency and generation quality. I am thus satisfied with the changes and recommend the paper for acceptance.

---

### Review · Reviewer_54ps · 2025-11-04

**Summary Of Contributions:**

This paper introduces PredLDM, a latent diffusion model approach for the spatiotemporal sequence prediction tasks. The core idea is a two-stage framework: first, a masked-attention transformer VAE (MT-VAE) compresses input video frames into a latent representation, capturing complex global spatiotemporal patterns; second, a condition-aware latent diffusion model (CA-LDM) generates future frames in latent space while explicitly modeling the given historical frames for better context understanding. In other words, the method leverages transformers with masked modeling to encode global coherence in the latent space and employs a diffusion process conditioned on past frames to ensure a comprehensive understanding of the history. The authors evaluate PredLDM on three diverse datasets (driving scenes from Kitti-Caltech, human actions from KTH, and weather radar data from SEVIR), demonstrating that it produces more realistic and accurate future frame predictions than a range of baseline models.

**Additional Comments:**

Additionally, I have some questions that could be clarified or further explored.

- **(Q1) Inference speed and efficiency.** Can the authors provide more details on PredLDM’s inference process and speed? Specifically, how many diffusion time steps are used to generate the future frames, and what is the typical runtime for predicting? This information is crucial for understanding the practicality of the model. Any quantitative comparison of inference time against a baseline would be very informative.

- **(Q2) Balancing conditional reconstruction vs. generation.** In the CA-LDM stage, the model optimizes a combination of two objectives – the standard diffusion loss for predicting future latent frames and the loss for reconstructing the condition frames. How are these two losses weighted or balanced during training? The paper mentions a “linear combination” but does not detail whether a specific coefficient or schedule is used. Moreover, did the authors observe any trade-off when adding the condition reconstruction loss?

- **(Q3) Scalability to higher resolution or longer sequences.** The experiments in the paper are conducted on sequences of relatively low resolution (e.g., 128$^2$ pixels) and predict a fairly short horizon (e.g., 10 frames). How well would PredLDM scale if we require higher-resolution predictions or a longer sequence of future frames?

**Audience:**

Yes

**Audience Explanation:**

The researchers in spatiotemporal prediction and video generative modeling will value a clear latent‑diffusion baseline with masked‑attention compression. The ablations and the condition‑aware objective provide actionable insights for building conditional diffusion predictors in practice.

**Broader Impact Concerns:**

No serious ethical or broader-impact concerns are apparent for this work. The technique is applied to relatively benign domains (driving scenes, human activities, weather patterns) and is aimed at improving predictive accuracy and realism, which can be beneficial in applications. One point to consider is that if such predictive models are used in safety-critical settings, reliability and uncertainty estimation become important. The authors might consider adding a discussion on uncertainty or failure cases in a camera-ready version to ensure users of PredLDM are aware of its limitations.

**Claims And Evidence:**

Yes

**Claims Explanation:**

There are the following strengths of this manuscript.

- **(S1) Interesting and practical design:** The proposed PredLDM sensibly combines modern generative modeling techniques. Using a transformer-based VAE with masked self-attention allows the model to learn a rich latent space encoding of video frames, overcoming the locality of pure convolutional encoders. The condition-aware diffusion process is cleverly formulated: during diffusion, the model not only learns to predict future frames from noise but also to reconstruct the conditioning frames in latent space.

- **(S2) Competitive comparison performances:** Firstly, the paper’s experimental section is comprehensive and well-designed to validate the claims. PredLDM demonstrates SOTA or competitive results on all evaluated datasets, indicating the method’s effectiveness. It consistently outperforms classical video prediction models and even recent diffusion-based models (MCVD, PVDM) in most metrics. Meanwhile, the model excels in the perceptual quality of predictions. Another impressive aspect is the temporal consistency of PredLDM’s outputs: the authors show that as one predicts further into the future, the degradation in PredLDM’s prediction quality is more gradual than for other models. Moreover, the paper includes extensive ablation studies and analyses.

- **(S3) Presentation:** The manuscript is generally well-written and organized, which is easy to follow. The inclusion of a pipeline diagram (Figure 1) and other visualizations (e.g., qualitative examples, error heatmaps) assists in understanding the model architecture and results. Also, the paper adheres to a professional structure and uses clear notation. Apart from a few minor language issues (addressed below), the writing quality is sufficient for conveying the complex methodology.

**Requested Changes:**

There are the following weaknesses of this manuscript, which could be clarified or improved during the rebuttal period.

* **(W1) Practical efficiency is under‑analyzed for a forecasting method.** Training uses frozen 2D‑VAE enc/dec, learned temporal transformer, a large DiT denoiser (N=32, hidden size 1152), and T=1000 sampling steps on 4 GPUs; yet there is no report of runtime, throughput, memory, or step ablations, which limits deployment insights.

* **(W2) Innovation is largely compositional rather than conceptual.** The approach combines masked modeling + transformer compression with latent diffusion and cross‑attention conditioning—all well‑established pieces. The “condition‑aware reconstruction” is a neat regularizer, but the paper does not position it against closely related diffusion‑for‑prediction works (e.g., MCVD, PVDM) beyond numeric gains, leaving the core novelty somewhat incremental.

* **(W3) Limited scale (resolution/horizon) and missing robustness.** All experiments run at 128×128 with 10‑step horizons; there is no test at higher resolutions or longer horizons, and no analysis of robustness (e.g., noise). This reduces evidence for applicability in realistic settings.

* **(W4)** Minor issues of writing. The manuscript is generally clear but has colloquial phrasing and minor typos; a careful pass is needed, e.g., wording around LPIPS highlights. Explanations of “condition‑aware reconstruction” could be tightened with a small schematic or pseudo‑code in the main text.

* **(W5) Additional issue after rebuttal.** I suggest the authors provide the classical metrics (i.e., MSE and MAE) used in previous video prediction works (e.g., OpenSTL benchmark). The current manuscript evaluates the proposed KVD and FVD with PSNR, SSIM, and LPIPS. These metrics are comprehensive but might not be friendly to previous video prediction methods like PredRNN and SimVP variants. Meanwhile, the authors could also add probabilistic evaluation and calibration analysis, especially for risk-sensitive domains. Given that diffusion models are inherently probabilistic and some target domains (e.g., weather nowcasting, autonomous driving) are safety-critical, it would be informative to quantify uncertainty quality.

---

> ### Author Response · Authors · 2025-11-28
> **Respond to Review of Paper5927 by Reviewer 54ps**
>
> **(W1 & Q1)** Exploring inference speed and efficiency.
>
> **Respond**: Thank you for your valuable suggestions on supplementing
> the experimental results on the cost-related evaluation of our work.
> Following your instruction, we have compiled the cost-related scores
> like inference time, FLOPs, model scale and training time compared to
> some baselines. A subsection '**A.6 Analysis on Computational Cost**' is
> added.
>
> - We report scores of ours and some baselines for better showing the
>   cost as in below Table. The results show that the model scale and
>   computational cost of classical predictive methods like ConvLSTM and
>   PredRNN are attractive. But these two cost-related scores of ours are
>   not very expensive compared to the diffusion baseline. Our inference
>   speed is faster than MCVD, processing one sequence in 1.33 seconds.
>   For training our architecture, firstly the MT-VAE is trained from
>   scratch with around 39.3 GPU hours, around 9 hours by 4 Nvidia GeForce
>   4090 GPUs. For the diffusion process, i.e., the second training stage
>   may cost 530.4 GPU hours, around 5.5 days with 4 Nvidia GeForce 4090
>   GPUs. When inferencing one sequence, 14.8 GB GPU memory is needed, it
>   can be easily deployed on Nvidia GeForce 4090. Considering the
>   compiled data here, we think the current setting is not quite
>   real-time, where PredLDM produces 10 frames in 1.331 seconds, even
>   with the compression ability from MT-VAE. If considering real-time
>   application, acceleration skills are still needed.
>
> - Added content in subsection subsection '**4.9 Discussion and
>   Limitation**', as 'Real-time performance. While our model demonstrates
>   competitive performance, it is important to acknowledge its
>   computational limitations. Although the model scale and computational
>   cost are not excessive compared to the diffusion baseline (MCVD) and
>   our inference speed of 1.33 seconds per sequence is faster, the
>   current setting is not ideal yet for real-world fast applications
>   satisfying real-time usage such as autonomous driving. Specifically,
>   generating 10 frames takes 1.331 seconds, even with the compression
>   provided by the MT-VAE. Therefore, for practical real-time deployment,
>   further acceleration techniques would be necessary. For offline usages
>   like urban supervision, analyzing or weather forecasting, it may
>   remain capable of serving.'
>
> - Besides, we are pleased to clarify that the denoising steps are set
>   1000 in our experiments, details are kindly added as in '**A.1 More
>   Details on Data and Implementation**'.
>
> **Table. Computational cost on FLOPS, model parameters, inference time,
> training time and GPU memory.**
>
>  |      |        FLOPs (G)  |   Params (M)  |  Inference time (s/seq)  |  Training time (GPU hours)  |   GPU memory (GB)|
>   |:-------:|:----------:|:-------:|:-------:|:-------:|:-------:|
>    |ConvLSTM   | 129.64   |  2.02   |    0.052   |   9.2 (1) / 2.3 (4)  |    2.1|
>   | PredRNN     |243.06  |   7.25     |  0.495    |   84.1 (1) / 21.0 (4) |     4.3|
>    |  MCVD    | 5,434.28 |  367.60    |  1.620  |    646.1 (1) / 161.5 (4) |    8.9|
>    |  Ours   |  4,566.57  | 421.26   |   1.331   |   530.4 (1) / 132.6 (4) |   14.8|

---

> > ### Author Response · Authors · 2025-11-28
> > **Respond to Review of Paper5927 by Reviewer 54ps - Continued Part 2**
> >
> > **(W2)** Adding discussion on condition-aware diffusion against
> > diffusion‑for‑prediction works.
> >
> > **Respond**: Thank you for your valuable suggestions and we are pleased
> > to clarify more on our work compared to existing diffusion predictive
> > works and make modifications by adding more discussion on our work to
> > enhance the novelty of our paper to the community. Following your
> > inspiring instructions, in the modified version, discussion below is
> > added. Firstly, we are pleased to discuss more on the masked attention
> > in MT-VAE, related with masking methods compared to DiT \[1\], SVD
> > \[2\], Make-A-Video \[3\] and MCVD \[4\]. DiT \[1\] proposes two kinds
> > of masking strategy noted by Only Video and Temporal Mask. For the first
> > masking, they use simple masking from 0 or 1 on time tokens. For the
> > second masking, they use frame-level one. Masking modeling is operated
> > on the data space, on temporal tokens or the direct pixel of frames. SVD
> > \[2\] additionally exploits a binary mask where 1 indicates the presence
> > of a conditioning frame and 0 refers to a mask embedding on the input of
> > U-Net denoising network. We can see their masking is on the input of
> > denoising network by concatenating a masking conditioning frame-size
> > embedding. Make-A-Video \[3\] uses the image-based masking trained for
> > frame interpolation, their masking is on data-level. MCVD \[4\] masks
> > frames in two ways including masking future/past frames, masking both
> > past and future frames. It directly masks historical frames or future
> > frames in data space. Different from these methods, our MT-VAE is
> > operated on the attention matrix, especially on the scaled dot
> > production matrix from query and key in transformers of VAE. Secondly,
> > more discussion is conducted on CA-LDM. From many classical generation
> > or prediction methods \[1,2,3,4\], they can all refer to the main
> > architectures DDPM or DDIM. For ours, architectural novelty may be the
> > CA-LDM by additional condition-aware reconstruction of historical
> > observations. Another diffusion denoising network is also used to
> > probabilistically modeling only on the condition, i.e., historical
> > spatiotemporal sequences. Our diffusion part design consists both the
> > commonly used conditional generation objection and importantly the
> > probabilistic modeling of the condition itself. The reason we design
> > this process in the denoising diffusion process is that different from
> > existing T2V research, the condition of this task is raw historical
> > spatiotemporal sequences while T2V methods use text as conditions, where
> > text is high dimensional and our condition is more low-level and with
> > plentiful structural details, more difficult to comprehend. So an
> > additional diffusion denoising process is designed in our PredLDM.
> >
> > 1: Xin Ma, et al. Latte: Latent diffusion transformer for video
> > generation. arXiv preprint arXiv:2401.03048 (2024).
> >
> > 2: Blattmann et al. Stable video diffusion: Scaling latent video
> > diffusion models to large datasets. arXiv preprint arXiv:2311.15127
> > (2023).
> >
> > 3: Singer et al. Make-a-video: Text-to-video generation without
> > text-video data. arXiv preprint arXiv:2209.14792 (2022).
> >
> > 4: Voleti et al. Mcvd-masked conditional video diffusion for prediction,
> > generation, and interpolation. Advances in neural information processing
> > systems 35 (2022), 23371--23385.
> >
> > **(Q2)** Dealing with weighting balance.
> >
> > **Respond**: Thank you for your professional suggestions and we are
> > pleased to clarify that the linear combination of two objectives (i.e.,
> > the standard diffusion loss and the condition-aware loss) is weighted by
> > 1:1.
> >
> > - We have made this more specific in the corresponding place of
> >   descriptions by 'In latent space, the denoising model is trained to
> >   predict less noisy samples by the 1:1 linear combination of ...' in
> >   '**A.1 More Details on Data and Implementation**'.
> >
> > - Besides, following your inspiring instructions, we have conducted
> >   additional experiments to investigate the trade-off between them.
> >   Added experimental results and discussion are placed in the revised
> >   manuscript. In details, results are reported in the table below. The
> >   setting for the conditional generation must not be 0, as this loss is
> >   critical to the task future prediction, so the minimum setting of this
> >   loss is 0.1. From experimental results on KittiCaltech, it appears
> >   that the setting of condition-aware loss is necessary. When the weight
> >   of $L_{CA}$ becomes larger, scores on KittiClatech are more
> >   competitive. In another side, the setting of $L_{LDM}$ should not be
> >   very small as this is the basic objective in spatiotemporal predictive
> >   tasks. It indicates that the linear combination of these loss
> >   functions is satisfied with the performance gain for common occasions.
> >
> > - A subsection named '**A.5 Sensitivity Analysis on Weighting Scheme of
> >   Diffusion-based Loss**' is added in the supplementary part of the
> >   modified version.

---

> > > ### Author Response · Authors · 2025-11-28
> > > **Respond to Review of Paper5927 by Reviewer 54ps - Continued Part 3**
> > >
> > > Table. Sensitivity analysis on the weighting scheme between conditional
> > > generation loss and condition-aware reconstruction loss on KittiCaltech.
> > >
> > >
> > > | $L_{LDM}$  | $L_{CA}$  | MSE   |       PSNR    |     SSIM       |  LPIPS|
> > > |:-------:|:----------:|:-------:|:-------:|:-------:|:-------:|
> > > |  0.1   |        0.9    |      0.169    |    19.18    |    0.606   |     0.252|
> > >  | 0.25    |      0.75     |    0.132   |     19.75   |     0.649     |   0.120|
> > >   |0.5     |      0.5     |     0.092    |    19.86    |    0.653    |    0.107|
> > >   |0.75    |      0.25    |     0.138    |    19.68    |    0.641     |   0.176|
> > >   |1       |      0     |       0.155  |      19.65     |   0.622     |   0.184|
> > >
> > > **(W3 & Q3)** Scalability to higher resolution or longer sequences, as
> > > well as adaptation to robustness.
> > >
> > > **Respond**: Thank you for your inspiring advice in designing more solid
> > > experiments for evaluating our PredLDM in longer horizons and higher
> > > resolutions. Following your professional instructions, we have conducted
> > > additional experiments to study related influence. A subsection named by
> > > '**A.7 Inference by Higher Resolutions and Longer Horizons'** is added
> > > in the modified version. To evaluate by higher resolutions and longer
> > > horizons, we adopt an iterative inference strategy. Given by 10 input
> > > frames, the model predicts the next 10 frames, which are then
> > > recursively fed back as input for subsequent predictions, as illustrated
> > > in Figure 11. Experimental results on the Sthv2 dataset (which is an
> > > added dataset) where frames are processed at 256\*256, revealing an
> > > acceptable decline in visual quality over extended time horizons, which
> > > is corroborated by a gradual degradation in quantitative metrics,
> > > including MSE, PSNR, SSIM, and LPIPS. For predicting the variable length
> > > of output sequences, all shorter predictions can directly be accessed
> > > from results in longer sequences, while longer horizons are available by
> > > this iterative inference strategy.
> > >
> > > **(W4) **Solving minor issues of writing.
> > >
> > > **Respond**: Thank you for your valuable suggestions. We have made
> > > modifications on writing and the pseudo-code is added.
> > >
> > > - For wording around LPIPS highlights, we have made a serious of
> > >   modifications, e.g., 'The most shining metric is LPIPS which resembles
> > >   the perceptual evaluating abilities similar to human, where 30.5%
> > >   improvement is achieved by PredLDM on KittiCaltech and 11.8%
> > >   improvement is made on KTH. Results on weather nowcasting show the
> > >   large margin in scores of PredLDM over other models.' in experimental
> > >   discussion in '**4.2 Results with Comparison to Existing Models**'.
> > >
> > > - Besides, we have added the pseudo-code for better explaining the
> > >   training process of condition-aware reconstruction in the subsection
> > >   '**3.3 Condition-aware Latent Diffusion**'.
> > >
> > > Overall, we hope our effort may help. Best wishes.

---

> > > > ### Comment · Reviewer_54ps · 2025-12-07
> > > > **Rebuttal Feedback**
> > > >
> > > > I would like to thank the authors for the detailed and constructive rebuttal and for the substantial revisions to the manuscript. Many of my earlier concerns have been addressed to some extent. Several additions (e.g., computational cost analysis, the clarification of the CA‑LDM loss) significantly improve the clarity and usability of the work. That said, there remain a few points where I still see room for clarification or further strengthening, which I commented on as follows.
> > > >
> > > > * **Conceptual novelty and relation to prior diffusion‑based predictors (W2).**
> > > > The revised manuscript and rebuttal provide a more elaborate discussion of how MT‑VAE and CA‑LDM differ from DiT, Stable Video Diffusion, Make‑A‑Video, and MCVD, in particular emphasizing that: masking is applied on the attention matrix rather than on frames or temporal tokens in input space; and CA‑LDM adds a condition‑aware reconstruction branch that models $p(c)$ in addition to $p(z|c)$, to improve the understanding of historical observations. This certainly helps to position the method architecturally. Still, from a conceptual standpoint, I feel that the novelty remains somewhat under‑substantiated:
> > > >
> > > >     * The benefits of attention‑matrix masking are demonstrated mostly through component‑wise ablations (e.g., Table 2 and the masking‑ratio study), but there is no end‑to‑end comparison against a “non‑masked transformer VAE + standard conditional LDM” full baseline.
> > > >
> > > >     * The claim that CA‑LDM maximizes “mutual information beyond the joint distribution” is largely conceptual; there is no quantitative evidence in terms of mutual information estimates, calibration metrics, or uncertainty measures that would distinguish CA‑LDM from more standard conditional diffusion predictors such as MCVD/PVDM.
> > > >
> > > >     * The relation to existing diffusion‑for‑prediction models is still primarily discussed at an architectural level (“we have an extra branch for p(c)”), without clearly demonstrating a qualitatively different behavior (e.g., better calibrated uncertainty, improved robustness to ambiguous histories, etc.).
> > > >
> > > >     I am fine with the fact that PredLDM may be more of a well‑engineered and strong baseline than a fundamentally new generative principle, but if the authors wish to emphasize CA‑LDM and masked attention as conceptually novel contributions, it would help to provide a clearer full‑system baseline (non‑masked transformer VAE + single‑branch conditional LDM), and discuss more concretely (possibly with one additional experiment) what behavioral advantage CA‑LDM brings beyond metric gains (e.g., in terms of diversity, calibration, or failure modes).
> > > >
> > > > * **Scalability and robustness (W3 / Q3).**
> > > > The new experiments on Sthv2 at $256\times 256$ with iterative roll‑out are a welcome addition, and they show that PredLDM can indeed operate at higher resolution and over longer horizons, with a gradual degradation of frame‑level metrics. Two aspects of my original concern remain only partially addressed:
> > > >     * Relative scalability on more complex datasets. On Sthv2, only the proposed model is evaluated, and no popular baselines are reported. As a result, it is hard to know whether PredLDM is relatively more scalable than existing predictors when data become more complex and horizons longer, or whether all methods degrade similarly.
> > > >
> > > >     * Robustness to noise / distribution shift. In W3, I explicitly mentioned robustness (e.g., to noise) as a missing aspect. The revised version still does not include. For example, any experiments with noisy, partially masked, or corrupted historical frames, or any explicit robustness‑oriented metrics. Even a small‑scale study (e.g., adding Gaussian noise or random masking to the input history, or testing mild domain shift on one dataset) would substantially strengthen the paper’s claims about reliability in realistic settings.
> > > >
> > > > Additionally, there are several new suggestions for the current manuscript.
> > > >
> > > > * I suggest the authors provide the classical metrics (i.e., MSE and MAE) used in previous video prediction works (e.g., OpenSTL benchmark). The current manuscript evaluates the proposed KVD and FVD with PSNR, SSIM, and LPIPS. These metrics are comprehensive but might not be friendly to previous video prediction methods like PredRNN and SimVP variants. Meanwhile, the authors could also add probabilistic evaluation and calibration analysis, especially for risk-sensitive domains. Given that diffusion models are inherently probabilistic and some target domains (e.g., weather nowcasting, autonomous driving) are safety-critical, it would be informative to quantify uncertainty quality.
> > > >
> > > > * For revision formatting, I strongly recommend highlighting all newly added or substantially modified text in a distinct color (e.g., \textcolor{blue}{...} in LaTeX) throughout both the main paper and the appendix, and briefly noting this convention in a footnote on the first page. This will greatly facilitate cross-checking of changes for reviewers.

---

> ### Author Response · Authors · 2025-12-17
> **Respond to Review of Paper5927 by Reviewer 54ps**
>
> We would like to thank the reviewer once again for your thorough and constructive feedback well with acknowledging revisions made to the manuscript. We sincerely appreciate your precious time and care you have taken to improve our work, providing more valuable suggestions which are critical to our current work and even future research. Your latest comments are insightful and will undoubtedly help us further strengthen the paper.
>
> We fully agree with your comments and have made a serious of modifications with additional experiments to respond to your suggestions. Following your professional instructions, four important parts are added or updated in this submission, including
>
> - 1) **Updated Figure 11** on experiments by longer horizons, mainly containing other competitive baselines such as the diffusion baseline PVDM and deterministic one SimVPv2. Updated captions and modified text are **highlighted by blue** as suggested using \textcolor{blue}{...} in LaTeX.
>
> - 2) Added experiments on robustness analysis for real-world applications as suggested are presented in **added Figure 12**, and an additional subsection is **added 'Robustness Analysis by Adding Noise'**. We evaluate the results by adding noise with different levels of standard deviation at 0.00, 0.05, 0.10 and 0.20 as your professional instruction.
>
> - 3) **Additional results on MSE/MAE** are provided as in **added Table 6**.
>
> - 4) Following your constructive suggestions, **uncertainty quality** is quantified as in **added Figure 13** and and additional subsection is added by **'Additional Results on MSE/MAE and Quantified Uncertainty'**. We plot the predictive standard variance at ensemble predictions, indicating higher predictability and lower intrinsic uncertainty for specific forecast cases.
>
> From these encouraging comments and our modifications, we can see evident progress on our work. In the future, we are going to consider more effort on quantifying the uncertainty of probabilistic models and conduct more calibration experiments for robust real-world usage as suggested by your valuable devotion. Best wishes.

---

### Review · Reviewer_Wx32 · 2025-11-16

**Summary Of Contributions:**

This paper presents PredLDM, a video prediction model based on latent diffusion models (LDMs). Specifically, they improve the current practices of the two-stage framework---(1) VAE training and (2) diffusion model training as follows. For (1), the authors propose MT-VAE, which uses random latents masking during training, enabling the latent space to better capture temporal dynamics compared with naive VAE training. They also add transformer blocks between the CNN encoder and decoder to achieve it. For (2), the authors propose to learn both the conditional generation and condition-aware reconstruction, which also models p(c) as well as the conditional distribution p(z|c). The authors argue that this strategy enables the model to comprehensively understand the temporal dynamics in the condition.

**Audience:**

Yes

**Audience Explanation:**

- I think jointly modeling p(c) along with p(z|c) would be something new, and some readers might be interested.
- It might be somehow related to recent diffusion-forcing and similar approaches.

**Claims And Evidence:**

Yes

**Claims Explanation:**

(Strengths)
- Overall, the paper is generally well-written and easy to follow.
- The proposed components (e.g., masking and condition reconstruction) are validated through ablation studies.
- The paper tries to conduct the claim on various datasets, e.g., KittiCaltech, KTH, and SEVIR.
- The paper tries to include many baselines.

(Weaknesses)
- The paper only deals with fine-grained datasets and does not include results on more complex datasets such as UCF-101 and Kinetics, which have been commonly used datasets in video prediction literature. I think adding these results is important to show the scalability of the proposed method with respect to the data complexity.
 - The paper deals with relatively short videos and low resolution: 128x128, and the target is to predict 10 frames. It would be great if the proposed method could be scalable to longer and higher-resolution video sequences.
- By the definition of a model, the model should be able to predict longer videos by conditioning the model on previously predicted video frames.  I believe trying this would greatly improve the quality of videos.

**Requested Changes:**

I would request the authors to add the following changes:
- Results on more complex datasets such as UCF-101 and Kinetics.
 - Higher resolution (at least 256x256) and longer video sequences (at least predicting 16 frames).
- The qualitative/quantitative results of the model predicting longer sequences with rollout
- Figure 2: I recommend that the authors add a legend globally, not just in each plot.

---

> ### Author Response · Authors · 2025-11-28
> **Respond to Review of Paper5927 by Reviewer Wx32**
>
> **(Weakness 1 & Requested Change 1)** Scalability to more complex
> datasets.
>
> **Respond**: Thank you for your inspiring suggestion in designing more
> solid experiments for evaluating our PredLDM adapting towards more
> complex datasets. Following your professional instructions, we have
> conducted additional experiments on a more complex dataset Sthv2. The
> Sthv2 dataset is notably complex due to its fine-grained action classes,
> where subtle differences in human-object interactions define distinct
> categories. It captures a vast range daily activities with high
> diversity. This makes it a challenging benchmark for predictive
> learning. From Figure 11b-11c, visualizations and results show that
> PredLDM can also deal well with more complex occasions.
>
> **(Weakness 2 & Requested Change 2-3)** Scalability to longer and
> higher-resolution video sequences.
>
> **Respond**: Thank you for your professional advice for evaluating our
> PredLDM in longer horizons and higher resolutions. Following your
> professional instructions, we have conducted additional experiments to
> study related influence. A subsection named by '**A.7 Inference by Higher
> Resolutions and Longer Horizons**' is added in the modified version. To
> evaluate by higher resolutions and longer horizons, we adopt an
> iterative inference strategy. Given by 10 input frames, the model
> predicts the next 10 frames, which are then recursively fed back as
> input for subsequent predictions, as illustrated in Figure 11.
> Experimental results on the Sthv2 dataset (which is an added dataset)
> where frames are processed at 256\*256, revealing an acceptable decline
> in visual quality over extended time horizons, which is corroborated by
> a gradual degradation in quantitative metrics, including MSE, PSNR,
> SSIM, and LPIPS. For predicting the variable length of output sequences,
> all shorter predictions can directly be accessed from results in longer
> sequences, while longer horizons are available by this iterative
> inference strategy.

---

> ### Author Response · Authors · 2025-11-28
> **Respond to Review of Paper5927 by Reviewer Wx32 - Continued Part 2**
>
> **(Weakness 3)** Taking suggestions on predicting longer sequences by
> conditioning the model on previously predicted video frames.
>
> **Respond**: Thank you for your inspiring suggestion on providing an
> encouraging manner for our model to predict at longer horizons. Inspired
> by this suggestion, we design the inferencing process as in Figure 11 by
> predicting more forward time sequences after feeding last predicted
> outcomes. Based on this strategy, we compile the corresponding
> experimental results as in the subsection '**A.7 Inference by Higher
> Resolutions and Longer Horizons**'.
>
> **(Requested Change 4)** Adding a legend globally for better display.
>
> **Respond**: Thank you for your professional suggestion and we are
> pleased to make corresponding modifications by replacing the legends per
> subfigure into a unified one. The modified figure is updated in the
> submitted modified version of our manuscript.
>
> Overall, we hope our effort may help. Best wishes.

---

> ### Comment · Reviewer_Wx32 · 2025-12-17
> **Response**
>
> Thanks for your response. I am convinced with the additional experiments. I do not have remaining concerns.

---

### Decision · Action_Editor_FYwy · 2025-12-19

**Recommendation:** Accept as is

**Audience:**

Yes

**Audience Explanation:**

Yes, the paper should be of interest to the video generation and predictive learning communities.

**Claims And Evidence:**

Yes

**Claims Explanation:**

I recommend acceptance of this manuscript. PredLDM presents a technically sound framework for spatiotemporal sequence prediction that combines masked-attention transformer-based VAEs with condition-aware latent diffusion models. While the approach is compositional rather than introducing fundamentally new principles, the paper demonstrates strong empirical results across diverse datasets, particularly excelling in perceptual quality metrics and temporal consistency.

All three reviewers converged on acceptance after the authors made substantial revisions to address their concerns: they added experiments on higher-resolution and longer-horizon prediction, provided detailed computational cost analysis (FLOPs, parameters, training time, inference latency), clarified the masked attention mechanism with precise mathematical formulation, included critical ablation studies (non-masked transformer VAE baseline, loss weighting sensitivity), supplemented robustness analysis with noise perturbation experiments, and enriched the discussion with insights on the distortion-perception tradeoff. The manuscript now provides a well-documented, rigorously evaluated baseline that will be of significant value to the video generation and predictive learning communities, meeting TMLR's standards for technically correct and reproducible research with clear methodology and accurate supporting evidence.